# Self-mixing in microtubule-kinesin active fluid from nonuniform to uniform distribution of activity

Teagan E. Bate[1], Megan E. Varney[2], Ezra H. Taylor[1], Joshua H. Dickie[1], Chih-Che Chueh [3], Michael M. Norton [4] & Kun-Ta Wu[1,5,6] ✉

Active fluids have applications in micromixing, but little is known about the mixing kinematics of systems with spatiotemporally-varying activity. To investigate, UV-activated caged ATP is used to activate controlled regions of microtubule-kinesin active fluid and the mixing process is observed with fluorescent tracers and molecular dyes. At low Péclet numbers (diffusive transport), the active-inactive interface progresses toward the inactive area in a diffusion-like manner that is described by a simple model combining diffusion with Michaelis-Menten kinetics. At high Péclet numbers (convective transport), the active-inactive interface progresses in a superdiffusion-like manner that is qualitatively captured by an active-fluid hydrodynamic model coupled to ATP transport. Results show that active fluid mixing involves complex coupling between distribution of active stress and active transport of ATP and reduces mixing time for suspended components with decreased impact of initial component distribution. This work will inform application of active fluids to promote micromixing in microfluidic devices.

Miniaturization enhances production efficiency in chemical engineering, biological engineering, and pharmaceutical manufacturing[1]. For example, microreactors—millimeter-scale devices with channels to mix chemicals and induce chemical reactions—are used to synthesize materials[2], test enzymes[3], and analyze protein conformations[4]. These devices require mixing to homogenize reactants, which is challenging because fluid dynamics at the micron scale are dominated by laminar flow. Mixing at a macroscopic scale is achieved by turbulence-induced advection repeatedly stretching and folding components until a uniform state is reached[5], but at a microscopic scale, turbulence is inhibited (Reynolds number ≪1) and mixing is dominated by molecular diffusion, which is slow and difficult to control. Approaches such as serpentine design[6] and vibrating bubbles[7] have been developed to enhance micromixing, but these are driven by external energy sources and thus require external components that limit miniaturization[1,8].

Active fluids—fluids with microscopic constituents that consume local fuel to generate movement[9–18]—have the potential to enhance mixing at the micron scale. Active fluids self-organize into chaotic turbulence-like flows[19–23] that promote micromixing by repeatedly stretching and folding fluid, even at low Reynolds numbers[24]. Prior work on active mixing has focused on active systems with uniform activity distribution[24–26]. However, mixing processes often start from a state of nonuniformity. Nonuniform distributions of activity in active fluid systems can cause complex dynamics[27–34]. Spatiotemporal patterns of activity that are prescribed from an external source[31,32] or emerge as an additional dynamical variable that coevolves with the system[27–29] have been studied. However, the effect of nonuniform distributions of activity on mixing has not been elucidated.

Here, we study the mixing dynamics of a microtubule-kinesin suspension whose activity is governed by the transport of ATP, the

[1]Department of Physics, Worcester Polytechnic Institute, Worcester, MA 01609, USA. [2]Department of Physics, New York University, New York, NY 10003, USA. [3]Department of Aeronautics and Astronautics, National Cheng Kung University, Tainan 701, Taiwan. [4]School of Physics and Astronomy, Rochester Institute of Technology, Rochester, NY 14623, USA. [5]Department of Mechanical Engineering, Worcester Polytechnic Institute, Worcester, MA 01609, USA. [6]The Martin Fisher School of Physics, Brandeis University, Waltham, MA 02454, USA. ✉e-mail: kwu@wpi.edu

system's energy source. We control the initial distribution of ATP by using caged ATP that can only fuel the fluid after exposure to ultraviolet (UV) light. This allows us to repeatedly observe the transient dynamics that carry the system from heterogeneous activity to homogeneous activity. We explore mixing dynamics ranging from diffusion-dominated to convection-dominated by varying the ATP concentration, composition of kinesin motors, flow cell geometry, and initial distribution of ATP. We contextualize the results with models at two levels of complexity. A simple model captures the mixing dynamics in a diffusion-limited regime, whereas a more complex model that includes active-fluid hydrodynamics reproduces aspects of observed enhanced transport and activity-dependent progression of the active-inactive interface.

## Results

### Self-mixing of active and inactive fluids

For the experiments presented herein, we select a 3D microtubule-kinesin active fluid because it enhances micromixing[14,24], has tunable activity[35–40], and has established models describing its flow behaviors[41–44]. In microtubule-kinesin active fluid, microtubules self-assemble by depletion into bundles that extend spontaneously, driving chaotic vortical flows. The extension is driven by kinesin motor

dimers that hydrolyze ATP to walk along pairs of antiparallel microtubules and force them in opposite directions (Fig. 1a)[14]. We augment the microtubule-kinesin system with UV light-activated chemistry that allows us to create distinct patterns of activity. In this light-activated system, the ATP is caged—its terminal phosphate is esterified with a blocking group (Fig. 1b)—such that it cannot be hydrolyzed by kinesin motors until the blocking group is removed by exposure to UV light[45,46]. In this system, the activation of the fluid is irreversible. After the fluid is activated, the action of the kinesin motors causes the microtubule network to become a 3D self-rearranging isotropic active gel consisting of extensile microtubule bundles that buckle and anneal repeatedly until the ATP is exhausted[14]. To quantify the evolution of the activity distribution, we suspend fluorescent tracers in the solvent and monitored the tracer motion to extract the speed distribution of active fluid flows (Fig. 1d). To observe the structure of the active suspension, we label microtubules with Alexa 647 (Fig. 1c).

When the fluid is in its inactive state, before it has been activated by UV light, the kinesin motor dimers are bound to microtubules, creating a quiescent crosslinked microtubule network that behaves like an elastic gel (Fig. 1c, top panel). The inactive gel is essentially isotropic, but after the fluid is loaded into a rectangular flow cell $(20 \times 4 \times 0.1\ \text{mm}^3)$ we observe some alignment of the bundles near

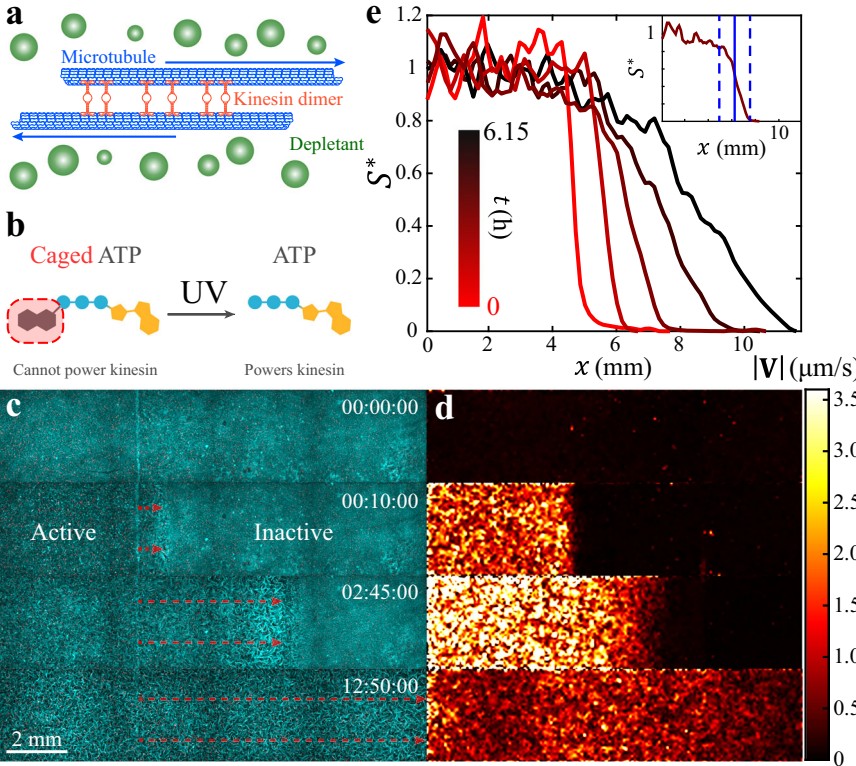

**Fig. 1 | (Experimental results) Mixing of activated and inactive microtubule-kinesin fluid. a** Microscopic dynamics in microtubule-kinesin active fluid. Depletants force microtubules into bundles where the microtubules can be bridged by kinesin motor dimers. The kinesin motors walk along the microtubules, forcing them to slide apart. The collective sliding dynamics cause the microtubules to form an extensile microtubule network that stirs the surrounding solvents and causes millimeter-scale chaotic flows[14]. **b** To develop an experimental system where we can create a distinct boundary between active and inactive fluid, we synthesize microtubule-kinesin active fluid with caged ATP. The caged ATP is not hydrolysable by kinesin motors, and thus cannot power the active fluid, until it is released by exposure to ultraviolet light[45,46]. This process is not reversible. **c** We expose only one side of the sample to ultraviolet light, which releases the ATP and activates the microtubule-kinesin mixture on that side of the channel. The released ATP

disperses toward the unexposed region, which activates the inactive fluid and expands the active region until the system reaches an activity-homogeneous state (Supplementary Movie 1). Because of the limited speed of multi-position imaging, only one-quarter of the active region is imaged. **d** Tracking tracer particles reveals the speed distribution of fluid flows, showing the activation of the left-hand side by UV light and the expansion of the active region into the inactive region. **e** Binning the same-time speeds vertically across the interface of active and inactive fluids reveals the speed profile $S$ which is normalized as $S^*(x) \equiv [S(x) - s_{in}]/[s_a - s_{in}]$, where $s_a$ is the average of speed profiles in the active zone and $s_{in}$ is the average of speed profiles in the inactive zone. Inset: The interface of the active and inactive fluids is determined as the region where the normalized speed profile is between 0.2 and 0.8 (dashed blue lines). The position of the interface is determined as where the normalized speed profile is 0.5 (solid blue line).

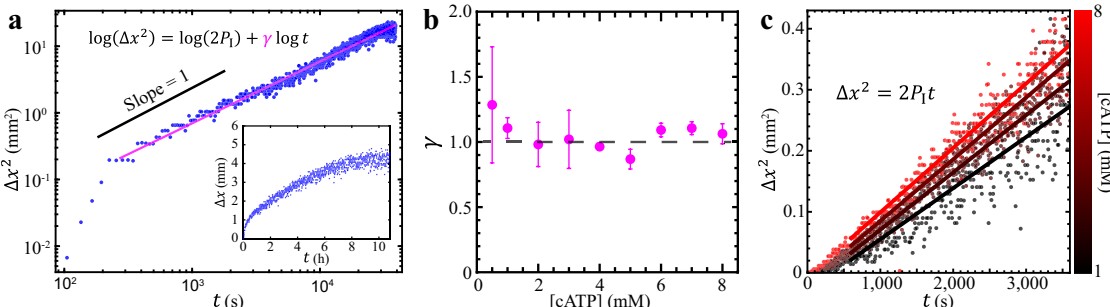

**Fig. 2 | (Experimental results) The progression of the active-inactive interface is governed by a diffusion-like process of ATP at the interface. a** Squared interface displacement ($\Delta x^2$) as a function of time ($t$) reveals a long-term ($t \gtrsim 200$ s) linear relation: $\Delta x^2 \sim t^\gamma$ with the interface progression exponent $\gamma \approx 1$. Inset: The interface displacement versus time shows that the interface moves rapidly initially and then gradually slows down. **b** The interface progression exponent is $\gamma \approx 1$ on average and is independent of the caged ATP concentration. Each error bar represents the standard deviation of $\geq 3$ trials. **c** Selected examples of squared interface displacement versus time for four caged ATP concentrations from 1 mM (black) to 8 mM (red). The progression rate of the interface is characterized with an interface progression coefficient $P_I$ determined by fitting the $\Delta x^2$ vs. $t$ data to $\Delta x^2 = 2P_I t$ with $P_I$ as the fitting parameter (colored lines). Increasing caged ATP concentrations increases the interface progression coefficient (steeper fit lines from black to red), which indicates that the interface progresses more quickly at higher caged ATP concentrations.

the boundary (Supplementary Fig. 1)[47]. To create an active-inactive interface, we use a mask to apply UV light to one side of the sample, which releases the ATP and activates the fluid on that side only (Fig. 1c, second panel; Supplementary Movie 1). The spatial pattern of activity evolves from a sharp interface to become increasingly diffuse as the initially active region invades the inactive region (Fig. 1d, second and third panels). We quantify this evolution of activity distribution with the normalized speed profile (Fig. 1e), which shows how the interface between regions widens and shifts as the active and inactive parts of the microtubule system blend over a period of hours.

## Characterization of the mixing dynamics

Active fluids enhance the motion of suspended tracers from diffusive (having a mean squared displacement [MSD] proportional to time lapse: MSD ~ $\Delta t^a$ with $a = 1$) to superdiffusive ($a > 1$)[14,26,35]. The progression of the active-inactive interface can also be described as diffusion-like or superdiffusion-like as follows: Suppose the displacement of the active-inactive interface is $\Delta x$ and the squared interface displacement increases with time as $\Delta x^2 \sim 2P_I t^\gamma$ with the interface progression coefficient $P_I$ and the interface progression exponent $\gamma$. If $\gamma = 1$, the progression of the active-inactive interface is defined as diffusion-like; if $\gamma > 1$, the progression of the active-inactive interface is defined as superdiffusion-like.

Because active fluids enhance microscale transport, we hypothesize that the active-inactive interface would progress in a superdiffusion-like manner ($\gamma > 1$). To test this hypothesis, we quantify the displacement of the active-inactive fluid interface as a function of time (Fig. 2a inset) and find that motion of the interface decelerates as the active fluid mixes with the inactive fluid such that the squared interface displacement progresses as $\Delta x^2 \sim t^\gamma$ with an interface progression exponent $\gamma \approx 1$ (Fig. 2a). We repeat the $\gamma$ measurement for caged ATP concentrations from 0.5 to 8 mM (0.5 mM is enough to maximize the flow speed of active fluid[48]) and consistently find that $\gamma \approx 1$ across this range (Fig. 2b). These results invalidate our hypothesis and suggest that the progression of active-inactive interface is diffusion-like. Notably, while the diffusive time-scaling $\gamma$ remains consistent, the prefactor $P_I$ exhibits a monotonic but nonlinear dependence on caged ATP concentration (Fig. 2c).

## Modeling with Fick's law and Michaelis-Menten kinetics

In the experiments, the interface progression coefficient $\gamma \approx 1$, which suggests that diffusion dominates the dynamics of the active-inactive interface. To contextualize our observations, we construct a minimal

model that combines diffusion of ATP with a previously measured relation between ATP concentration and local fluid velocity[48]. Herein, we model ATP's dispersion using Fick's law of diffusion:

$$\frac{\partial C(\boldsymbol{r}, t)}{\partial t} = D\nabla^2 C(\boldsymbol{r}, t), \tag{1}$$

where $C(\boldsymbol{r}, t)$ represents the spatial distribution of ATP concentrations at time $t$ and $D$ is the diffusion coefficient of ATP in active fluid. We choose $D = 140 \ \mu\text{m}^2/\text{s}$, which is one-fifth the diffusion coefficient of ATP in water[49], because the crosslinked microtubule network makes the fluid more viscous than water[50] and our measurement on diffusion coefficient of suspended fluorescein is one-fifth of its reported value in aqueous solution (Supplementary Note 1). To simplify the modeling, we consider a 1D active fluid system confined in a segment, $x = 0 - L$, where $L = 20$ mm is the segment length, and apply no-flux boundary conditions

$$\frac{\partial}{\partial x} C(x=0, t) = \frac{\partial}{\partial x} C(x=L, t) = 0. \tag{2}$$

To mimic the UV-activation process (Fig. 1c), we initiate the ATP concentration with a step function

$$C(x, t=0) = \frac{C_0}{2} \text{erfc}\left(\frac{x - x_0}{2\sqrt{\epsilon D}}\right), \tag{3}$$

where $C_0 = 0.5 - 8$ mM, the initial ATP concentrations in the activated region; erfc is the complementary error function; and $x_0 = 10$ mm is the initial position of the active-inactive interface. We choose $\epsilon = 0.001$ to generate a sharp concentration transition at the interface. We numerically solve Eqs. (1–3) to determine the spatial and temporal distribution of ATP concentrations (Fig. 3a; Supplementary Movie 2). To relate the evolving ATP distribution to local flow speed, we leverage previous experimental results[48] that find the average velocity in bulk samples follows Michaelis-Menten kinetics

$$\bar{v}(x, t) = \bar{v}_\text{m}\left[\frac{C(x, t)}{C(x, t) + K}\right], \tag{4}$$

where $\bar{v}$ is the mean speed of active fluid, $\bar{v}_\text{m} = 6.2 \ \mu\text{m/s}$ is the saturated mean speed, and $K = 270 \ \mu\text{M}$ is the ATP concentration that leads to half of the saturated mean speed, $\bar{v}_\text{m}/2$. (Model selection is described in Supplementary Note 2). The mean speed distributions

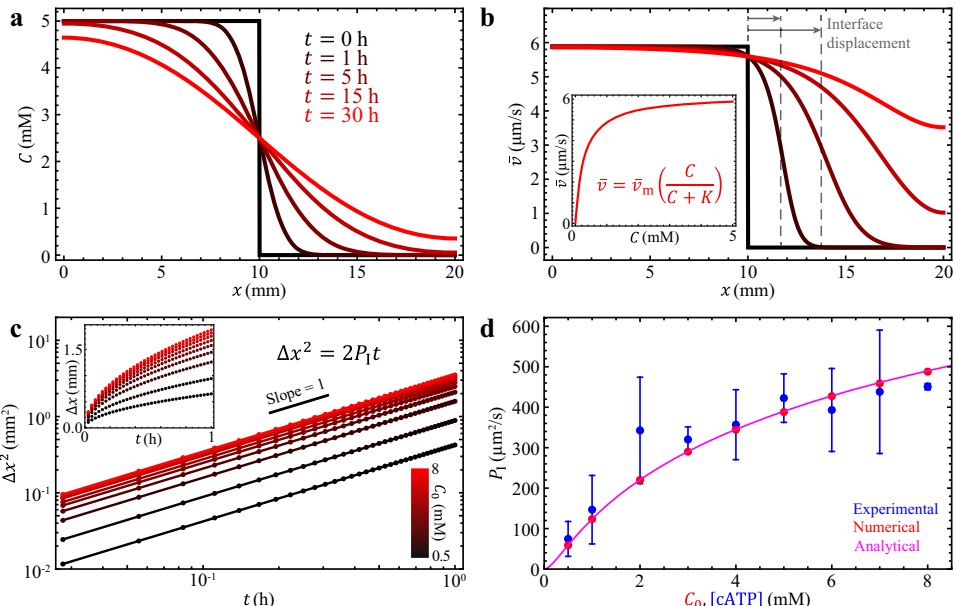

**Fig. 3 | (Modeling results) Fick's law of diffusion and Michaelis-Menten kinetics captures the diffusion-like mixing of active and inactive fluids. a** The simulated distribution of ATP concentrations starts as a step function (black, $t = 0$ h) and then develops into a smoothed hill function (red, $t = 30$ h) as ATP evolves from a one-sided distribution to a homogeneous state. **b** The model converts the ATP distribution into the speed distribution of active fluid via Michaelis-Menten kinetics: $\bar{v} = \bar{v}_\mathrm{m}[C/(C + K)]$, where $\bar{v}_\mathrm{m} = 6.2\ \mu m/s$ and $K = 270\ \mu M$ (based on our previous studies[48]). The corresponding mean speed distribution of active fluid evolves from a step function distribution (black, $t = 0$ h) to a near-constant function (red, $t = 30$ h) (Supplementary Movie 2). Inset: The plot of the Michaelis-Menten equation (Eq. 4). **c** In the simulation, the diffusion-driven mixing process leads the squared interface displacement to be proportional to time, regardless of initial ATP

concentration $C_0$ (see Supplementary Note 3 for derivation of $\Delta x^2 \propto t$). Inset: Interface displacement increases rapidly with time initially, followed by a gradual deceleration similar to the experimental observation (Fig. 2a inset). **d** In the simulation, the interface progression coefficient $P_\mathrm{I}$ is determined by fitting the $\Delta x^2$ vs. $t$ data (Panel **c**) to $\Delta x^2 = 2P_\mathrm{I}t$ with $P_\mathrm{I}$ as fitting parameter. The model $P_\mathrm{I}$ increases with the initial concentration of ATP, $C_0$ (red dots), similarly to how the experimentally analyzed $P_\mathrm{I}$ varies with caged ATP concentration (blue dots; each error bar represents the standard deviation of $\geq 3$ trials). The model $P_\mathrm{I}$ and experimental $P_\mathrm{I}$ differ by only -10%. The magenta curve shows the analytical solution, $P_\mathrm{I}(C_0)$ (Supplementary Equation 7), which reproduces the numerical results (red dots). (See Supplementary Note 3 for derivation of $P_\mathrm{I}$ as a function of $C_0$).

(Fig. 3b) show that initially one side of the sample is activated (black) and then the sample evolves toward a more uniformly activated state (red). The squared interface displacement of the active-inactive interface increases linearly with time, $\Delta x^2 \sim t$ (Fig. 3c; see Supplementary Note 3 for derivation of $\Delta x^2 \propto t$), which matches our experimental observation of $\gamma \approx 1$ (Fig. 2a, b). Further, we compare the dependency of the interface progression coefficient $P_\mathrm{I}$ (determined by fitting $\Delta x^2$ vs. $t$ data to $\Delta x^2 = 2P_\mathrm{I}t$ with $P_\mathrm{I}$ as the fitting parameter; Figs. 2c and 3c) on initial ATP concentrations between experiment and model and also find excellent agreement (differed by -10%; Fig. 3d). Taken together, the agreement between simulation and experiment on the scaling of the dynamics (Figs. 2b and 3c) and dependency on initial ATP concentration (Fig. 3d) indicate that the dispersion of ATP is dominated by diffusion and that Michaelis-Menten kinetics are appropriate for a coarse-grained model to connect ATP concentration with local flow speed of active fluid[48], without the need to introduce a more complex hydrodynamic model[42,51].

**Superdiffusion-like progression of active-inactive interface**
The success of the diffusion-limited model suggests that the active transport in the active fluid systems studied above is dominated by diffusion. This inspires us to question whether the progression of active-inactive interface will become superdiffusion-like when the active transport becomes convection-like[27,28]. To answer this question, we varied experimental parameters to explore a wider range of fluid flow speeds. To achieve lower flow speeds, we alter the composition of motor proteins by replacing a fraction of the processive motors (K401), which exert force on microtubules continuously, with non-processive motors (K365) that detach after each force application. The

reduced number of processive motors has the net effect of driving the extensile motion of microtubules more slowly (Fig. 4a inset left)[36,48]. To achieve higher flow speeds, we increase the height of the sample container to decrease hydrodynamic drag (Fig. 4a inset right)[48,52]. Throughout these experiments, we keep the caged ATP concentration constant (5 mM).

As in the previous experiments, we analyze the spatiotemporal progression of activity to find the interface progression exponent $\gamma$ as a function of the average flow speed in the bulk of the initially activated area, $\bar{v}_\mathrm{ab}$ (Fig. 4a). Because changing channel geometry alters the characteristic size of vortices in active fluids[39], we unify our datasets by plotting $\gamma$ as a function of the Péclet number, Pe (Fig. 4b), defined as Pe $\equiv \bar{v}_\mathrm{ab}l_\mathrm{c}/D$ where $l_\mathrm{c}$ is the correlation length of flow velocity (see Supplementary Note 4) and $D = 140\ \mu m^2/s$ is our estimate of ATP diffusion in the system (see Supplementary Note 1)[53,54]. The Péclet number is a dimensionless quantity representing the ratio of convective transport rate to diffusive transport rate. A larger Péclet number (typically of order 10 or above) indicates convection-dominated active transport, and a smaller Péclet number (typically of order 1 or below) indicates diffusion-dominated active transport. Our data show that for Pe $\lesssim 3$, the interface progression exponent remains $\gamma \approx 1$ (Fig. 4b), which corresponds to the regime captured by our model (Fig. 3). Then as Pe increases to greater than 3, $\gamma$ grows monotonically (Fig. 4b). For the largest Pe explored in our experiments (Pe $\approx 16$), $\gamma$ reaches -1.7, which indicates that convective processes are beginning to emerge and dominate the active transport. Overall, our data suggest that as the active transport transitions from diffusion-dominated (Pe $\lesssim 3$) to convection-dominated (Pe $\gtrsim 3$) regimes, the progression of active-inactive interfaces transitions from diffusion-like ($\gamma \approx 1$) to superdiffusion-like ($\gamma > 1$).

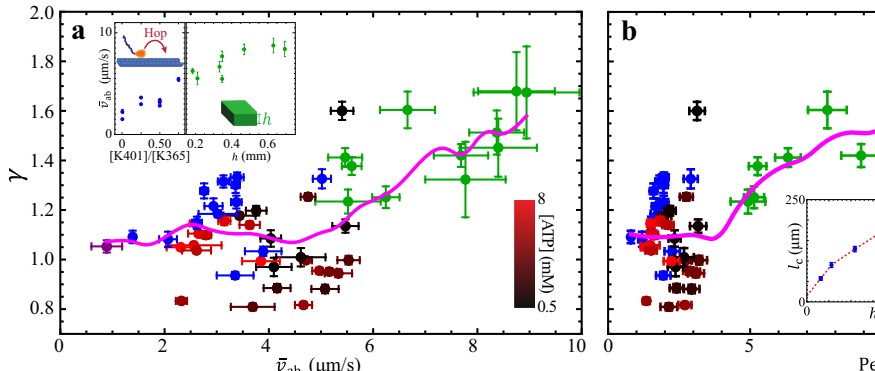

**Fig. 4 | (Experimental results) Transition of active-inactive interface progression.** Progression of the active-inactive interface transitions from diffusion-like ($\gamma \approx$ 1) to superdiffusion-like ($\gamma > 1$) as the active transport changes from diffusion-dominated (Pe$\lesssim$ 3) to convection-dominated (Pe$\gtrsim$ 3). **a** The active-inactive interface progression exponent ($\gamma$) increases with the flow-speed level of the active fluid ($\bar{v}_{ab}$). Shown are data from experiments with low ATP concentration (0.5 mM, black dots), high ATP concentration (8 mM, red dots), decreased flow speeds (from nonprocessive motors partially replacing processive motors; blue dots), increased flow speeds (from increased sample height; green dots)[48], and both nonprocessive motors and increased sample height (purple dot). The magenta curve represents the moving average of $\gamma$. Although the analyzed $\gamma$ from each experiment is noisy, the moving averaged $\gamma$ exhibits an overall monotonic increase with the flow-speed level of active fluid $\bar{v}_{ab}$. Each dot represents one experimental measurement. Each error bar in $\gamma$ represents the slope fitting error in $\ln\Delta x^2 = \ln(2P_1) + \gamma \ln t$ (Fig. 2a), and

each error bar in $\bar{v}_{ab}$ represents the standard deviation of flow speeds in the active region. Inset: The flow-speed level of the active fluid is tuned by replacing processive motors (K401) with nonprocessive motors (K365) with the same overall motor concentrations (120 nM) (left)[36,48] or by altering the sample height (right)[48,52]. **b** The same data as in Panel a, plotted as a function of Péclet number, Pe $\equiv \bar{v}_{ab}l_c/D$, where $l_c$ is the correlation length of flow velocity in active fluid deduced from sample container height $h$ (inset) and $D$ is the diffusion coefficient of ATP. Each error bar in $\gamma$ is the same as in Panel a, and each error bar in Pe represents propagated uncertainties from $\bar{v}_{ab}$ in Panel a and $l_c$ in inset. Inset: Correlation length of flow velocity in active fluid $l_c$ increases monotonically with sample container height $h$[39]. The red dashed line represents the line interpolation of blue dots. The error bars represent the standard deviations of two trials. (See Supplementary Note 4 for measurements and analyses of $l_c$).

## Dispersion of UV-activated fluorescent dyes

To this point, we have characterized the mixing of active and inactive fluids by the progression of the interface between them; however, like milk blending into coffee, the mixing process often involves dispersion of suspended components. To characterize how suspended components disperse during the progression of the active-inactive interface, we design another series of experiments with suspended components that are initially nonuniform. We dope inactive fluid with suspended UV-activated fluorescent dyes and expose one side of the sample container to UV light, which simultaneously activate the fluid and the fluorescent dye. We find that in an inactive sample ($\bar{v}_{ab} = 0$), where dyes disperse only by molecular diffusion, the dye barely disperse, whereas in a sample where one side is activated ($\bar{v}_{ab} = 8.2\,\mu$m/s), the dyes are transported by active fluid flows and almost completely disperse through the sample in 4 hours (Fig. 5a). To quantify the dispersion rate, we adopt Saintillan and Shelley's method[25] to analyze the normalized multiscale norm of dye brightness as a function of time: $\hat{s}(t) \equiv |s(t)|/|s(0)|$, where

$$|s| \equiv \left[ \sum_{\boldsymbol{k}} \frac{|s_{\boldsymbol{k}}|^2}{\sqrt{1 + l^2 k^2}} \right]^{1/2}, \quad (5)$$

$s_{\boldsymbol{k}}$ is the Fourier coefficient at wave vector $\boldsymbol{k}$ in a Fourier expansion of the dye brightness and $l = 4.84\,\mu$m is the pixel size of the micrographs. We find that the normalized multiscale norm decays faster as $\bar{v}_{ab}$ increases from 0 to 8.2 $\mu$m/s (Fig. 5b). In light of reports that the norm decays exponentially[25], we quantify the decay rate by fitting the first hour $\hat{s}(t)$ data to $\ln\hat{s} = -t/t_0$ with $t_0$ (mixing time) as the fitting parameter (Fig. 5b inset) and find that the mixing time decreases with flow speed of active fluid (Fig. 5c inset). When the fluid is inactive ($\bar{v}_{ab} = 0$), dye dispersion is dominated by molecular diffusion and the mixing time is 24 hours; slightly activating the fluid ($\bar{v}_{ab} = 2\,\mu$m/s) reduces the mixing time to 8 hours, which demonstrates that active fluid flows enhance the mixing process of suspended components[26].

To reveal how the mechanism of active transport (i.e., diffusion-dominated or convection-dominated) alters the mixing time, we analyze the mixing time as a function of the Péclet number and find that the mixing time monotonically decreases as the active transport becomes more convection-like (Fig. 5c). Notably, there is no discernible transition in mixing time as the active transport transitions from diffusion-dominated to convection-dominated, although there is a transition in the progression of active-inactive interfaces (Fig. 4b). This dependence of mixing time on the Péclet number in active-inactive fluid systems is similar to that in an activity-uniform active fluid system (Supplementary Note 5 and Supplementary Fig. 6b), which shows that Péclet number is the controlling parameter for mixing time of suspended components in active fluid systems, regardless of the distribution of activity.

## Continuous active fluid model

Our experimental data show that as the active transport becomes more convection-like, the active-inactive interface progression transitions from diffusion-like to superdiffusion-like (Fig. 4b) and the mixing time of suspended components decreases monotonically (Fig. 5c). To determine whether this complex mixing process can be modeled with an existing active fluid model, we adopted Varghese et al.'s model[51] because it successfully describes the transition from coherent to chaotic flow in 3D microtubule-kinesin active fluid systems[52]. The model describes microtubules as self-elongating rods whose nematic order, $\mathbf{Q}$, is subject to spontaneous decay due to the rods' rotational molecular diffusion and reorientation by solvent flow. Thus, the dimensionless kinetic equation for $\mathbf{Q}$ can be written as:

$$\partial_{t^*}\mathbf{Q} + \mathbf{u} \cdot \nabla_*\mathbf{Q} + \mathbf{Q} \cdot \boldsymbol{\Omega}^* - \boldsymbol{\Omega}^* \cdot \mathbf{Q} = -\mathbf{Q} + \nabla_*^2\mathbf{Q} + \lambda\left[\frac{2}{d}\mathbf{E}^* + \mathbf{Q} \cdot \mathbf{E}^* + \mathbf{E}^* \cdot \mathbf{Q} - \frac{2}{d}\text{Tr}\left(\mathbf{Q} \cdot \mathbf{E}^*\right)\mathbf{I}\right],$$
$$(6)$$

where $t^*$ is the dimensionless time, $\nabla_*$ is the dimensionless spatial gradient operator, $\nabla_*^2$ is the dimensionless Laplacian operator, $\boldsymbol{\Omega}^* \equiv [(\nabla_*\mathbf{u})^{\mathsf{T}} - \nabla_*\mathbf{u}]/2$ is the dimensionless vorticity tensor,

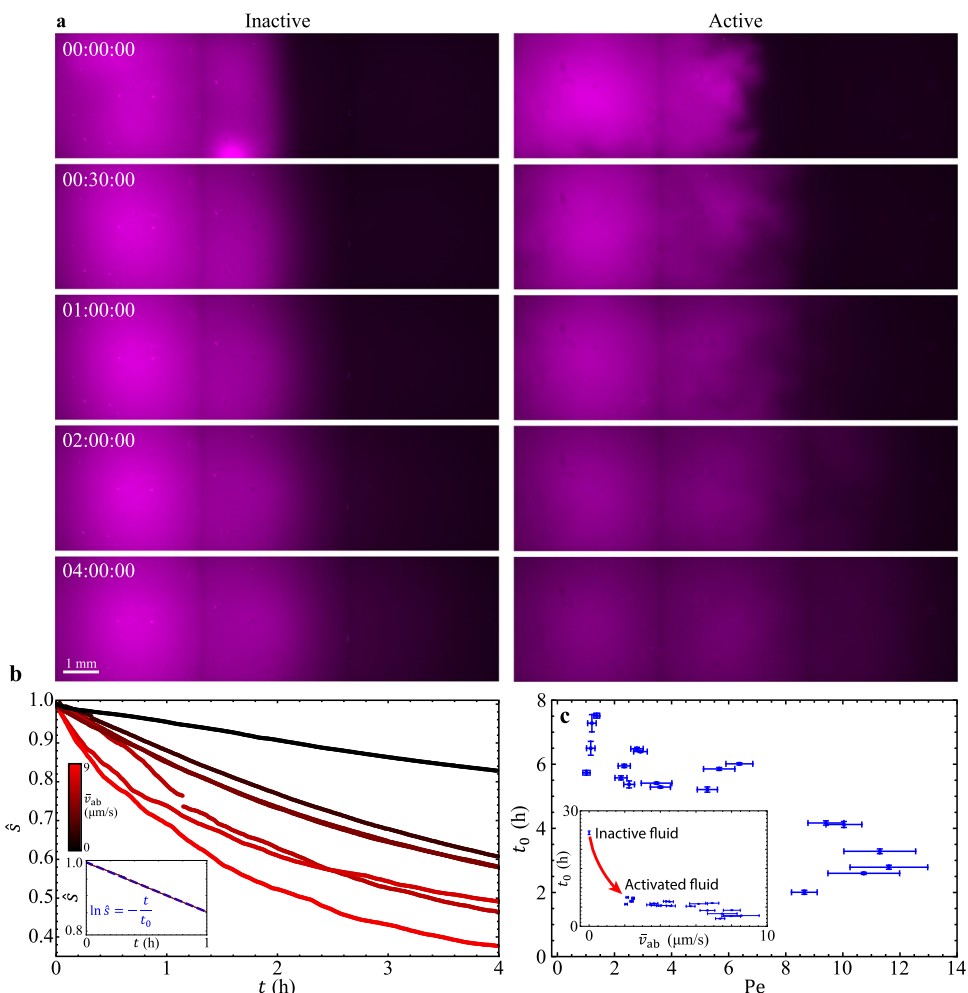

**Fig. 5 | (Experimental results) Dispersion of UV-activated fluorescent dyes in active-inactive fluid systems.** Active fluid flows promote mixing of UV-activated fluorescent dyes, which are initially activated in the left-hand side of the container only. **a** Dispersion of UV-activated fluorescent dyes (magenta) in inactive ($\bar{v}_{ab}$ = 0; left column) and active ($\bar{v}_{ab}$ = 8.2 μm/s; right column) microtubule-kinesin fluid. Active fluid flows actively transport fluorescent dyes and enhance their dispersion. Time stamps are hour:minute:second. (See also Supplementary Movie 3.) **b** Selected examples of normalized multiscale norm vs. time for different active bulk flow speeds, $\bar{v}_{ab}$. Normalized multiscale norm $\hat{s}(t)$ decreases faster in a faster-flowing active fluid system. Inset: The normalized multiscale norm, $\hat{s}(t)$, in log-linear axes behaves as a straight line, which suggests that the norm decays exponentially with time. The decay time scale $t_0$ (or mixing time) is determined by fitting the

normalized multiscale norm versus time data to $\ln\hat{s} = -t/t_0$ with $t_0$ as the fitting parameter (dashed blue line). **c** The mixing time decreases monotonically with Péclet number, which demonstrates that a stronger convection mechanism leads to faster mixing of suspended components. Each dot represents one experimental measurement. Each error bar in $t_0$ represents the slope fitting error in $\ln\hat{s} = -t/t_0$ (Panel **b** inset), and each error bar in Pe (defined as Pe $\equiv \bar{v}_{ab}l_c/D$) represents propagated uncertainties from $\bar{v}_{ab}$ (see inset) and $l_c$ (see Supplementary Fig. 5d). Inset: Mixing time, $t_0$, as a function of active bulk mean speed, $\bar{v}_{ab}$. Each error bar in $t_0$ is the same as in Panel **c**, and each error bar in $\bar{v}_{ab}$ represents the standard deviation of flow speeds in the active region. Notably, the mixing time of the inactive fluid system ($\bar{v}_{ab}$ = 0) is 24 hours (top-left dot); minimally activating the fluid ($\bar{v}_{ab}$ = 2 μm/s) reduces the mixing time to 8 hours.

$\mathbf{E}^* \equiv [(\mathbf{\nabla}_*\mathbf{u})^T + \mathbf{\nabla}_*\mathbf{u}]/2$ is the dimensionless strain rate tensor, $\lambda = 1$ is the flow alignment coefficient, and $d$ is the system dimensionality. The dimensionless flow field $\mathbf{u}$ is governed by the Stokes equation

$$\nabla_*^2\mathbf{u} - \mathbf{\nabla}_*p^* - \mathbf{\nabla}_* \cdot \boldsymbol{\sigma}_a = 0 \qquad (7)$$

and incompressibility constraint ($\mathbf{\nabla}_*\bullet\mathbf{u} = 0$), where $p^*$ is the dimensionless pressure and $\boldsymbol{\sigma}_a \equiv \alpha^*\mathbf{Q}$ is the dimensionless active stress exerted by self-elongating rods with a dimensionless activity coefficient $\alpha^*$[55]. Because the activity coefficient increases with ATP concentration[56], we select an $\alpha$-ATP relation[57]

$$\alpha^* = \alpha_0^* \frac{C}{C+K} , \qquad (8)$$

where $\alpha_0^*$ is the dimensionless activity level, $C$ is the ATP concentration, and $K$ = 270 μM[48]. We select this relation because it captures the

dynamics of microtubule bundle extension and kinesin kinetics (Michaelis-Menten), which play critical roles in the activity of microtubule-kinesin active fluid systems[58,59]. Finally, given that ATP diffuses as a result of thermal fluctuation as well as flows with the active fluid, we model ATP dispersion with a convection-diffusion equation:

$$\partial_{t^*}C = D^*\nabla_*^2C - \mathbf{u} \cdot \mathbf{\nabla}_*C, \qquad (9)$$

where $D^*$ is the dimensionless ATP molecular diffusion coefficient. To simplify modeling, we consider a 2D active fluid system ($d = 2$)[51] confined in a 112 × 22 rectangular boundary with no-slip boundary condition for flows ($\mathbf{u} = \boldsymbol{0}$) and no-flux boundary condition for rods ($\mathbf{n} \cdot \mathbf{\nabla}_*\mathbf{Q} = \boldsymbol{0}$, where $n$ represents a unit vector normal to boundaries). To solve the equations for $\mathbf{Q}$, $\mathbf{u}$, and $C$ (Eqs. 6, 7, and 9), we determine the initial conditions as quiescent solvent ($\mathbf{u} = \boldsymbol{0}$) under uniform pressure ($p^* = 0$) with the rods in an isotropic state [$Q_{xx} = -Q_{yy} = 2.5 \times 10^{-4}$ rn(**r**) and $Q_{xy} = Q_{yx} = 5 \times 10^{-4}$rn(**r**), where

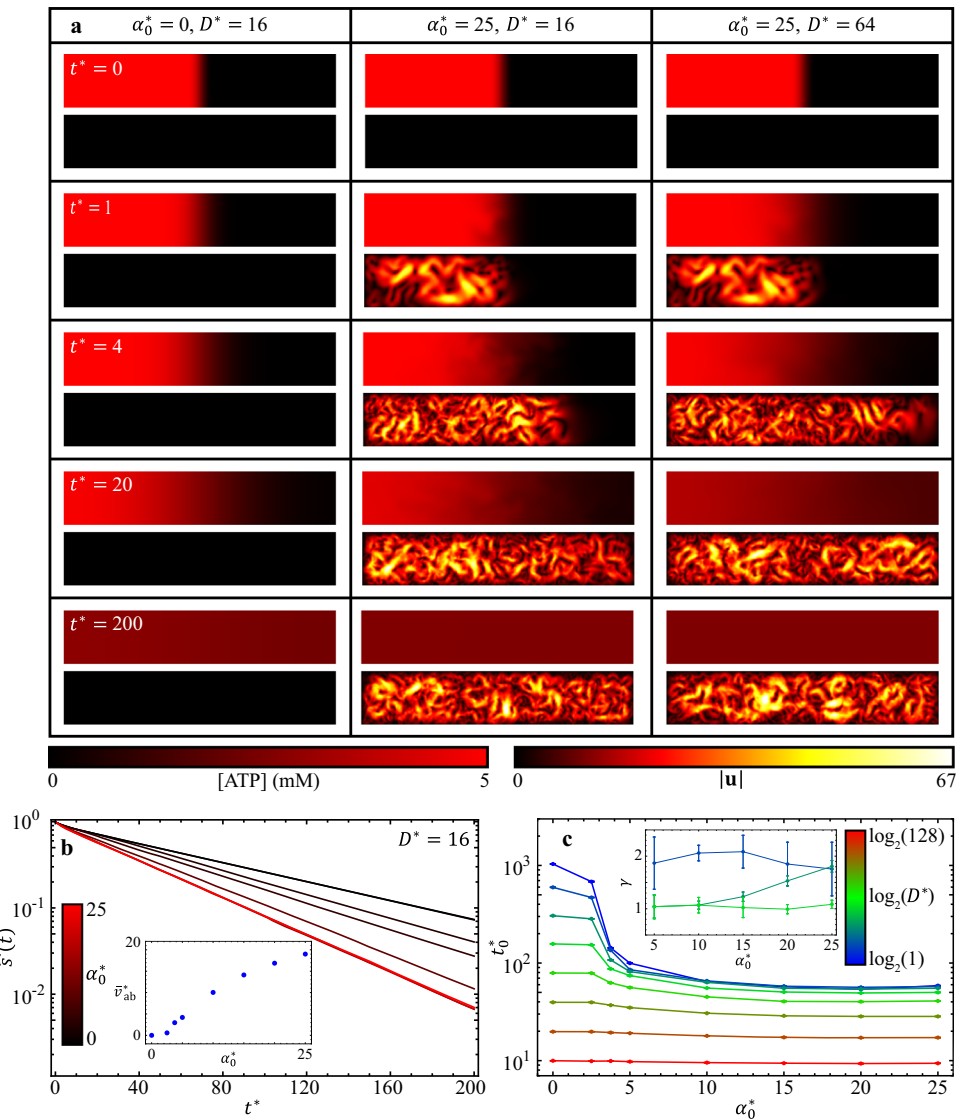

**Fig. 6 | (Modeling results) A continuous active fluid simulation reveals that the mixing time of ATP depends on the dimensionless molecular diffusion coefficient of ATP and the dimensionless activity level of active fluid. a** Table of ATP concentration (top panels) and active fluid flow speed (bottom panels) maps for various dimensionless activity levels $\alpha_0^*$ and molecular diffusion coefficients $D^*$. When the fluid has no activity ($\alpha_0^* = 0$; left column), ATP disperses to the right side of the system only by molecular diffusion; the dispersion is enhanced when the active fluid starts to flow and actively transport ATP ($\alpha_0^* = 25$; middle column). The dispersion is further enhanced when ATP diffuses significantly faster ($D^* = 64$; right column) (Supplementary Movie 4). **b** Evolution of normalized multiscale norm for $\alpha_0^* = 0$–25 while keeping $D^* = 16$. The normalized multiscale norms decay

exponentially with time: $\hat{s} = \exp(-t^*/t_0^*)$, where $t_0^*$ is the dimensionless mixing time. Inset: Dimensionless mean speed of active fluid in active region $\bar{v}_{ab}^*$ monotonically increases with dimensionless activity level $\alpha_0^*$. **c** Dimensionless ATP mixing times, $t_0^*$, as a function of dimensionless activity level, $\alpha_0^*$, for various dimensionless molecular diffusion coefficients, $D^*$. Increasing both $\alpha_0^*$ and $D^*$ decreases mixing time monotonically. Each error bar in $t_0^*$ represents the fitting error of $\hat{s}$ vs. $t^*$ to $\ln\hat{s} = -t^*/t_0^*$ (Panel **b**). Inset: Active-inactive interface progression exponent $\gamma$ as a function of dimensionless activity level $\alpha_0^*$ for dimensionless molecular diffusion coefficients $D^* = 2$ (dark blue), 4 (dark green), and 8 (light green). Each error bar in $\gamma$ represents the slope fitting error as in Fig. 2a. Increasing $D^*$ decreases $\gamma$ (from dark blue to light green curve), whereas increasing $\alpha_0^*$ increases $\gamma$ (dark green curve).

rn(**r**) is a spatially uniform random number between −1 and +1] and 5 mM of ATP distributed on only one side of the system. Then we evolve the fluid flows and ATP distributions for 200 units of dimensionless time ($t^* = 0$–200) with the finite element method[60].

Our modeling results (Supplementary Movie 4) show that in an inactive system ($\alpha_0^* = 0$; $D^* = 16$; Fig. 6a, left column), ATP disperses only by molecular diffusion, but when one side of the sample is activated ($\alpha_0^* = 25$; $D^* = 16$; middle column), the system develops chaotic turbulence-like mixing flows that actively transport the ATP toward the inactive region. In a third simulation where the ATP molecular diffusion rate is increased ($\alpha_0^* = 25$; $D^* = 64$; right column), the mixing process speeds up. These simulation results show that the process of

ATP dispersion is controlled by both molecular diffusion of ATP and active fluid-induced convection.

To quantify the efficacy of ATP mixing by active fluid, we analyze the normalized multiscale norm of ATP concentrations as a function of time[25] for $\alpha_0^* = 0$–25 and $D^* = 1$–128 (Eq. (5) with $l = 1$). We find that the norms decay exponentially with time: $\hat{s} \sim \exp(-t^*/t_0^*)$, where $t_0^*$ is the dimensionless mixing time (Fig. 6b), which is consistent with results reported by Saintillan and Shelley[25]. We analyze mixing time as a function of dimensionless activity level, $\alpha_0^*$, for each dimensionless molecular diffusion coefficient $D^*$ (Fig. 6c) and find that when ATP diffuses slowly ($D^* \lesssim 16$; blue to light green curves), mixing time decreases with increasing activity level or faster active transport (Fig. 6b inset), which is consistent with our experimental observation

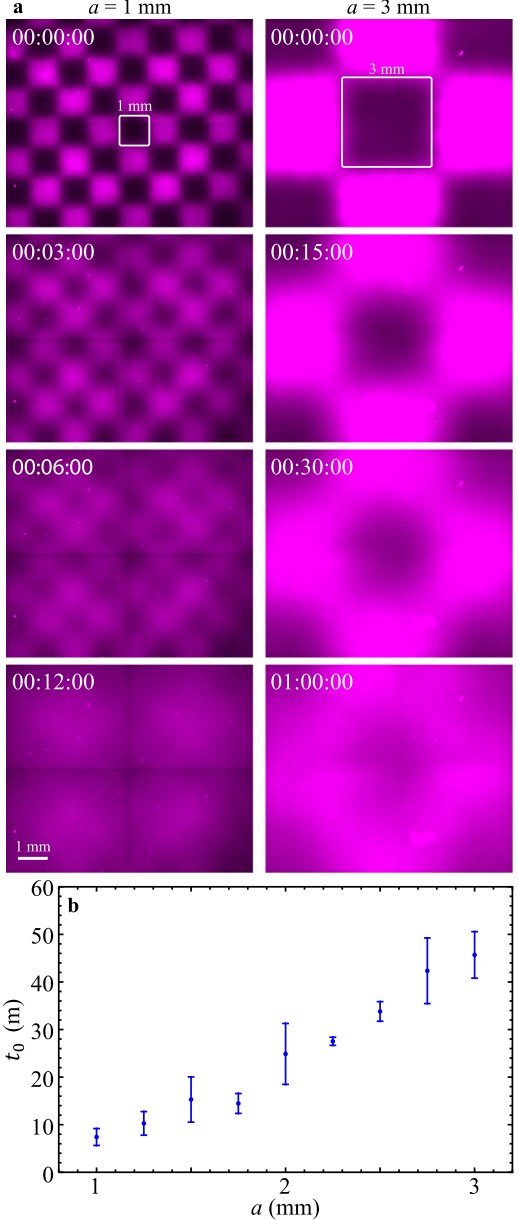

**Fig. 7 | (Experimental results) Fluid activated in a checkerboard pattern mixes faster when the checkerboard grid is smaller. a** Checkerboard-patterned UV lights are used to activate active fluid and caged fluorescent dyes. (Supplementary Movie 5). **b** The mixing time of the checkerboard-activated fluid increases with grid size, which demonstrates that the mixing efficacy of active fluid depends on distribution of activity: more nonuniform active fluid mixes the system more slowly. Each error bar in $t_0$ represents the standard deviation of two trials.

(Fig. 5c). Our simulation also shows that as ATP diffuses sufficiently fast ($D^* \gtrsim 32$; olive and red curves), the mixing time is nearly independent of activity level. Overall, increasing both the molecular diffusion coefficient and the activity level decreases mixing time. Thus, our simulation shows that both molecular diffusion (represented as $D^*$) and active fluid-induced convection (related to $\alpha_0^*$) play important roles to disperse and homogenize the suspended components; which mechanism dominates the dispersion depends upon the competition between these two mechanisms.

To demonstrate how the competition of these two mechanisms affects the progression of active-inactive interfaces, we analyze the interface progression exponent $\gamma$ as a function of $\alpha_0^*$ for various diffusion coefficients $D^*$ (Fig. 6c inset) and find that when the diffusion mechanism is relatively weak ($D^* = 2$; dark blue curve), the convection mechanism dominates the interface progression, leading it to progress in a superdiffusion-like, or more precisely, ballistic-like manner ($\gamma \approx 2$). Contrarily, as the diffusion mechanism becomes relatively strong ($D^* = 8$; light green curve), diffusion mechanisms dominate the interface progression, leading it to progress in a diffusion-like manner ($\gamma \approx 1$). Interestingly, we find that in an intermediate strength of diffusion mechanism ($D^* = 4$; dark green curve), increasing activity level $\alpha_0^*$ transitions the interface progression from diffusion-like to super-diffusion-like, which is consistent with our experimental observation (Fig. 4). Overall, our active-fluid hydrodynamic model qualitatively captures the mixing dynamics of active and inactive fluid systems in terms of active-inactive interface progression and dispersion of suspended components.

## Checkered distribution of dyes and activity

Up until this point we have only explored one configuration of non-uniform active fluid systems: an activated bulk on one side of a channel adjacent to an inactive bulk on the other side. To explore how other spatial configurations of activity affect mixing, we use a checkerboard pattern of UV light to split the activated region into cells. As in previous experiments, 50% of the total fluid is activated. Fluid activated in a checkerboard pattern evolves to a homogeneous state more quickly than fluid that is activated on one side only (1 hour vs. 10 hours; Fig. 7a). UV-activated fluorescent dyes show that the mixing time decreases as the grid size decreases from 3 mm to 1 mm (Fig. 7b).

To elucidate how checkerboard mixing driven by active fluid differs from that driven by molecular diffusion alone, we apply our established active-fluid hydrodynamic model for both active ($\alpha^* = 25$) and inactive ($\alpha^* = 0$) fluid systems (Fig. 8a; Supplementary Movie 6). As expected, we find that the mixing time increases monotonically with grid size for both active and inactive fluid systems (Fig. 8b), with the active fluid system (red curve) having a shorter mixing time than the inactive fluid system (black curve). Interestingly, we find that when the grid size is sufficiently small ($a \lesssim 5$), the active and inactive fluids have the same simulated mixing time. We also find that as the grid size increases from 5 to 22, the mixing time of the inactive fluid increases more than the mixing time of the active fluid (40× vs. 3×).

## Discussion

The self-mixing process of microtubule-kinesin active fluid with non-uniform activity is driven by active transport at the active-inactive interface. We estimate the contributions of diffusive and convective transport using the Péclet number, Pe. We find that when the active transport is dominated by the diffusion mechanism (Pe $\lesssim 3$), the active-inactive interface progresses in a diffusion-like manner ($\gamma \approx 1$; Fig. 2). These dynamics are quantitatively captured by a Fick's law-based model that quasi-statically related local activity to the local concentration of ATP by using a previously measured ATP-velocity relation (Fig. 3)[48].

As we raise the Péclet number (Pe $\gtrsim 3$) by increasing both the local fluid velocity and mixing length scale, we find that the active-inactive interface concomitantly progresses in a more superdiffusion-like manner ($\gamma > 1$; Fig. 4). We observe experimentally that increasing the Péclet number decreases the mixing time of suspended fluorescent dyes (Fig. 5c), which demonstrates that more convective flow mixes the suspended components faster. These results, along with the progression of the active-inactive interfaces, are qualitatively captured by an active-fluid hydrodynamic model (Fig. 6c) that couples active stress-induced fluid flow and transport of ATP molecules (Eqs. (6–9)).

Interestingly, while our hydrodynamic model predicts interface progression exponent $\gamma = 2$ for high activity levels (Fig. 6c inset), in our experiments $\gamma$ appears to plateau at $\gamma \approx 1.7$ (Fig. 4b). Our model may have overestimated $\gamma$ because the microtubule network in the inactive portion of the sample is crosslinked by immobile kinesin motor dimers

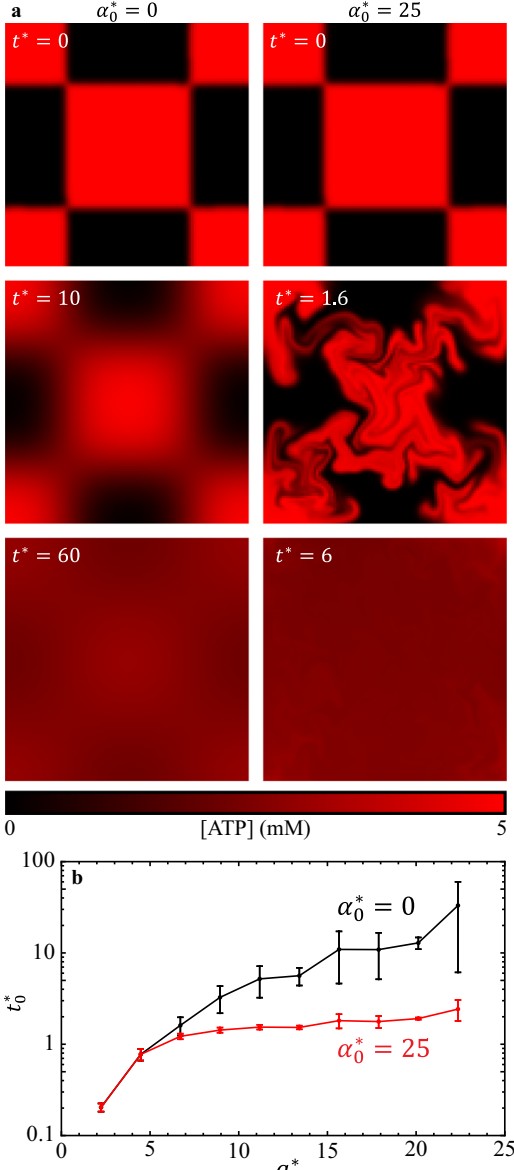

**Fig. 8 | (Modeling results) Simulations with ATP initially distributed in a checkerboard pattern in a 45 × 45 simulation box. a** Distributions of ATP (red) disperse from the checkerboard pattern (Eq. [11] with $a^* = 22$) to a homogeneous state by molecular diffusion only (left; $D^* = 1$, $\alpha_0^* = 0$) and by the combination of molecular diffusion and active fluid-induced convection (right; $D^* = 1$, $\alpha_0^* = 25$). **b** The dimensionless mixing time of ATP increases monotonically with the dimensionless checkerboard grid size $a^*$ in both active (red) and inactive (black) fluid systems. Error bars represent the standard deviation of two trials performed with different types of checkerboard pattern (Eq. ([11]) and ([12])).

that cause it to behave like an elastic gel. When the ATP molecules arrive at the active-inactive interface, they fuel the motor dimers, which fluidizes the network. Crucially, this fluidizing/melting process takes time to develop[47,50]. Thus, for the interface to progress, not only does ATP need to be transported to the inactive fluid region, but the ATP-fueled motors also need time to melt the gel-like microtubule network into a fluid. Such melting dynamics can delay the progression of the active-inactive interface and lower $\gamma$. In the simulation, the melting dynamics are absent; the network melts almost instantly as soon as ATP arrives at the inactive fluid, and $\gamma$ only depends on active transport of ATP. Our additional studies (Supplementary Note 6) support the idea of a network melting mechanism by showing that the progression of the active-inactive interface falls behind the

progression of ATP molecules (Supplementary Fig. 7e), whereas in the simulation the fronts of both coincide (Supplementary Fig. 8c). Future research to elucidate the network melting dynamics can involve monitoring dyes, tracers, and microtubules simultaneously to reveal the correlations among ATP dispersion, active fluid flows, and microtubule network structure (melting). The process can be modeled with the active-fluid hydrodynamic model used herein, modified to include ATP-dependent rheological constants and additional relevant dynamic processes to represent the melting process of the gel-like network at the interface.

We also find that the distribution of activity has a significant effect on mixing time. Systems consisting of more, smaller active areas (checkerboard pattern; high uniformity (Fig. [7]) evolve to a homogeneous state faster than systems with the same total active area distributed as one piece (one side active; low uniformity (Fig. [1]c). This is likely because the smaller grid size increases the active-inactive interface area, which allows the active fluid to interact with inactive fluid more efficiently. Interestingly, our active-fluid hydrodynamic model shows that when the grid size is sufficiently small, the mixing times of active and inactive fluids are indistinguishable (Fig. [8]b). This may be because the active fluid needs time to warm up from an initial quiescent state before reaching its steady activity state[47]. In experiments, the system has a warm-up time that may have been caused by network melting (Supplementary Note 6). Although a network melting mechanism is not included in the model, the simulated active fluid flow takes dimensionless time to rise because the onset of the flows is triggered by the initial activity-driven instability in extensile **Q** field, which takes finite dimensionless time to develop (~1 dimensionless time in this case; Supplementary Movie 6)[25,47]. Thus, in cases where the grid size is sufficiently small, the model shows molecular diffusion completing the mixing before emergence of active fluid flows. We also find that mixing time in an active fluid system is less sensitive to initial distribution of activity than that in an inactive system (Fig. [8]b), which suggests that introducing active fluid to a microfluidic system can drastically reduce the impact of the initial condition on mixing efficacy.

This study has limitations. Observations on the mixing of active microtubule-kinesin fluid and inactive microtubule-kinesin fluid may not be generalizable to cases in which active fluid mixes with other types of fluid. Also, we do not characterize the degree of chaos in the system, such as by measuring Lyapunov exponents and topological entropies[24]. Future research can track tracers in 3D and measure how these quantities change in the 3D isotropic active microtubule network at different strengths of active transport (i.e., Lyapunov exponent vs. Péclet number and topological entropy vs. Péclet number).

Another limitation of this study is that our results for interface progression transitioning from diffusion-like to superdiffusion-like (Fig. [4]) are based on large length-scale data that we analyze considering the interface as one piece with a specific position coordinate (Fig. [2]a). However, the interface is the region where ATP concentration decays from saturation (>$K$; see Eq. ([4])) to 0, and according to previous studies[14,35] tracer motion in this region should transition from superdiffusion-like to diffusion-like behaviors. Directly measuring the mean squared displacement of tracers across the active-inactive interface can elucidate the transition of the interface progression behaviors on the microscopic scale. Such measurements are not practical in our system because the active-inactive interface changes position and width with time (Fig. [1]e); tracers initially in the diffusive zone can later be in the superdiffusive zone as the interface passes by, and it will be difficult to distinguish between the diffusive and superdiffusive data. Future research to elucidate tracer behaviors at active-inactive interfaces can utilize fluid that is only active when it is exposed to light[31,32] to provide a stable activity gradient and thus obtain a reliable mean squared displacement of tracers at different parts of the interface.

Overall, this work demonstrates that mixing in nonuniform active fluid systems is fundamentally different from mixing in uniform active fluids. Mixing in nonuniform active fluid systems involves complex interplays among spatial distribution of ATP, active transport of ATP (which can be either diffusion-like or convection-like, depending on Péclet number), and a fluid-gel transition of the microtubule network at the interface[47]. This work paves the path to the design of microfluidic devices that use active fluid to promote or optimize the micromixing process[8] to enhance production efficiency in chemical and biological engineering and pharmaceutical development[1]. The results may also provide insight into intracellular mixing processes, because the cytoplasmic streaming that supports organelles within cells is powered by cytoskeletal filaments and motor proteins that function similarly to microtubule-kinesin active fluid[61].

## Methods

### Polymerize microtubules

Microtubules constitute the underlying network of microtubule-kinesin active fluid. Microtubules are polymerized from bovine brain α- and β-tubulin dimers purified by three cycles of polymerization and depolymerization[62,63]. The microtubules (8 mg/mL) are then stabilized with 600 μM guanosine-5′[(α,β)-methyleno]triphosphate (GMPCPP, Jena Biosciences, NU-4056) and 1 mM dithiothreitol (DTT, Fisher Scientific, AC165680050) in microtubule buffer (80 mM PIPES, 2 mM MgCl₂, 1 mM ethylene glycol-bis(β-aminoethyl ether)-N,N,N′,N′-tetraacetic acid, pH 6.8) and polymerized by a 30-minute incubation at 37 °C and a subsequent 6-hour annealing at room temperature before being snap frozen with liquid nitrogen and stored at −80 °C. The microtubules are then labeled with Alexa Fluor 647 (excitation: 650 nm; emission: 671 nm; Invitrogen, A-20006) and mixed with unlabeled microtubules at 3% labeling fraction during polymerization to image microtubules for non-fluorescein experiments (Figs. 1, 2, and 4). For fluorescein experiments, the microtubules are unlabeled (Figs. 5 and 7).

### Dimerize kinesin motor proteins

Kinesin motor proteins power the extensile motion of sliding microtubule bundle pairs in active fluid by forming a dimer and walking on adjacent antiparallel microtubules to force them in opposite directions (Fig. 1a)[14,64]. We express kinesin in the *Escherichia coli* derivative Rosetta 2 (DE3) pLysS cells (Novagen, 71403), which we transform with DNA plasmids from *Drosophila melanogaster* kinesin (DMK) genes[65]. For most experiments in this paper, we use processive motors that include DMK's first 401 N-terminal DNA codons (K401)[66]. To explore the effect of low mean speed of active fluid bulk on the interface progression exponent γ, we mix in fractions of nonprocessive motors whose plasmid includes DMK's first 365 codons (K365, Fig. 4 inset left)[36,48,67]. The kinesin motors are tagged with 6 histidines enabling purification via immobilized metal ion affinity chromatography with gravity nickel columns (GE Healthcare, 11003399). To slide adjacent microtubule bundle pairs, kinesin motors need to be dimerized, so the kinesin motors are tagged with a biotin carboxyl carrier protein at their N terminals, which allow the kinesins to be bound with biotin molecules (Alfa Aesar, A14207)[14,62]. To dimerize the kinesin, we mix either 1.5 μM K401 processive motors or 5.4 μM K365 nonprocessive motors with 1.8 μM streptavidin (Invitrogen, S-888) and 120 μM DTT in microtubule buffer, incubate them for 30 minutes at 4 °C, and then snap freeze them with liquid nitrogen and store them at −80 °C.

### Prepare microtubule-kinesin active fluid

To prepare the active fluid, we mix 1.3 mg/mL microtubules with 120 nM kinesin motor dimers and 0.8% polyethylene glycol (Sigma 81300), which acts as a depleting agent to bundle microtubules (Fig. 1a)[14]. Kinesin steps from the minus to the plus end of microtubules

by hydrolyzing ATP and producing adenosine diphosphate[59]. To control the initial spatial distribution of ATP and thus the activity distribution of active fluid, we use 0.5 to 8 mM caged ATP (adenosine 5′-triphosphate, P3-(1-(4,5-dimethoxy-2-nitrophenyl)ethyl) ester, disodium salt and DMNPE-caged ATP, Fisher Scientific, A1049), which is ATP whose terminal phosphate is esterified with a blocking group rendering it nonhydrolyzable by kinesin motors unless exposed to 360-nm UV light. Exposure to UV light removes the blocking group (Fig. 1b) and allows the kinesin motors to hydrolyze the ATP into ADP and activate the active fluid[45,46]. The ATP hydrolyzation decreases ATP concentrations, which slow down the kinesin stepping rate and thus decrease active fluid flow speed[14,35,48,58,59]. To maintain ATP concentrations so as to stabilize the activity level of the active fluid bulk over the course of our experiments, we include 2.8% v/v pyruvate kinase/lactate dehydrogenase (Sigma, P-0294), which converts ADP back to ATP[14,68]. To feed the pyruvate kinase enzyme, we add 26 mM phosphenol pyruvate (BeanTown Chemical, 129745). We image the active fluid samples with fluorescent microscopy for 1 to 16 hours, which can bleach the fluorescent dyes and thus decrease the image quality over the course of experiments. To reduce the photobleaching effect, we include 2 mM trolox (Sigma, 238813) and oxygen-scavenging enzymes consisting of 0.038 mg/mL catalase (Sigma, C40) and 0.22 mg/mL glucose oxidase (Sigma, G2133) and feed the enzymes with 3.3 mg/mL glucose (Sigma, G7528)[14]. To stabilize proteins in our active fluid system, we add 5.5 mM DTT. To track the fluid flows, we dope the active fluid with 0.0016% v/v fluorescent tracer particles (Alexa 488-labeled [excitation: 499 nm; emission: 520 nm] 3-μm polystyrene microspheres, Polyscience, 18861). To test how active fluid can mix suspended components, we introduce 0.5 to 6 μM caged, UV-activated fluorescent dyes (fluorescein bis-(5-carboxymethoxy-2-nitrobenzyl) ether, dipotassium salt; CMNB-caged fluorescein, ThermoFisher Scientific, F7103), which are colorless and nonfluorescent until exposed to 360-nm UV light[45]. The dye concentration is chosen to maintain a sufficient signal-to-noise ratio while avoiding brightness saturation in micrographs. Upon UV exposure, the fluorescein is uncaged and thus becomes fluorescent and can be observed with fluorescent microscopy. Because the fluorescent spectrum of the fluorescein overlaps with our Alexa 488 tracers, for our experiments with caged fluorescein (Fig. 5) we replace the tracers with Flash Red-labeled 2-μm polystyrene microspheres (Bangs Laboratories, FSFR005) and use unlabeled microtubules (0% labeling fraction) to prevent fluorescent interference from microtubules while imaging tracers.

### Prepare active-inactive fluid systems

To prepare the active-inactive fluid system, we load the inactive microtubule-kinesin fluid with caged ATP to a polyacrylamide-coated glass flow cell (20 × 4 × 0.1 mm³) with Parafilm (Cole-Parmer, EW-06720-40) as a spacer sandwiched between a cover slip (VWR, 48366-227) and slide (VWR, 75799-268)[36] and seal the channel with epoxy (Bob Smith Industries, BSI-201). Then we mask one side of the sample with a removable mask of opaque black tape (McMaster-Carr, 76455A21) attach to a transparent plastic sheet (Supplementary Fig. 9a) and shine UV light on the sample for 5 minutes before removing the mask (Supplementary Fig. 10). In the unmasked region, the UV light releases the ATP from the blocking group and activates the fluid by allowing the ATP to fuel the local kinesin motors; in the masked region, the fluid remains quiescent (Fig. 1c; Supplementary Movie 1)[45]. To explore how the progression exponent changes with active fluid bulk mean speed, we accelerate fluid flows by making the flow cell taller by stacking layers of Parafilm to decrease hydrodynamic resistance (Fig. 4a inset right)[48,52]. To explore how the spatial nonuniformity of activity influences the mixing efficacy of the active-inactive fluid system, we mask the sample with checkerboard-patterned masks (FineLine Imaging, Fig. 7a).

## Image samples with dual fluorescent channels

We image the active fluids with epifluorescent microscopy (Ti2-E Inverted Microscope, Nikon, MEA54000) with the commercial image acquisition software Nikon NIS Elements version 5.11.03. To capture a wide area of the sample ($20 \times 4$ mm$^2$), we use a $4\times$ objective lens (CFI Plan Apo Lambda $4\times$ Obj, Nikon, MRD00045, NA 0.2) to image 3 to 4 adjacent frames rapidly ($\lesssim 3$ s) and stitch the micrographs into one large image for flow and dye dispersion analyses (Figs. 1, 2, 4, 5, and 7; Supplementary Fig. 9b).

Performing these analyses requires monitoring at least two components in two fluorescent channels in each sample; for example, the dye dispersion experiments (Fig. 5) requires analyzing fluorescent dyes (excitation: 490 nm; emission: 525 nm) and Flash Red-labeled tracers (excitation: 660 nm; emission: 690 nm) simultaneously. This could have been accomplished by programming a microscope to rapidly switch back and forth between filter cubes, but this would have quickly worn down the turret motor and the time required to switch filter cubes ($\geq 4$ s) and move the stage to capture adjacent images and stitch them (~3 s) would have made the minimum time interval between frames $\geq 10$ s, which would have prevented us from tracking high-density tracers (1000 mm$^{-3}$ with a mean separation of 5 μm in a 0.1-mm-thick sample) whose speeds are 1 to 10 μm/s, even with a predictive Lagrangian tracking algorithm[69]. To overcome this technical challenge in imaging our samples, we establish a dual-channel imaging system that consisted of a multiband pass filter cube (Multi LED set, Chroma, 89402–ET) and voltage trigger (Nikon) placed between the light source (pE-300$^{\text{ultra}}$, CoolLED, BU0080) and camera (Andor Zyla, Nikon, ZYLA5.5-USB3). Instead of changing filter cubes, the multiband pass filter cube allows us to switch between multiple emission and excitation bands by switching between channels with the same filter cube. We alternatively activate the blue (401–500 nm) and red (500–700 nm) LEDs to excite and observe the fluorescent dyes and tracers almost simultaneously. The LED light source communicates directly with the camera via voltage triggering to coordinate LED activation time and bypass computer control to further boost the light switching rate. This technique shortens our channel switching time to 3 to 5 μs; thus the time interval between image acquisitions of different fluorescent channels is only limited by the exposure times of each channel. This setup allows us to image two fluorescent channels almost simultaneously (within milliseconds) and thus enables us to monitor two fluorescent components side-by-side, such as microtubules and tracers (Fig. 1c; Supplementary Movie 1), caged fluorescent dyes and tracers (Fig. 5c), and caged fluorescent dyes and microtubules (Supplementary Movie 5).

## Analyze positions of active-inactive fluid interfaces

We characterize the mixing kinematics of active and inactive fluids by analyzing the interface progression exponents $\gamma$ and coefficients $P_1$ (Figs. 2–4). These analyses require identification of the interface positions. We determine the interface positions by first tracking tracers in sequential images with the Lagrangian tracking algorithm[69], which reveals the tracers' trajectories $r_i(t)$ and corresponding instantaneous velocities $v_i \equiv d\mathbf{r}_i/dt$. Then we analyze the speed profile of tracers by binning the tracer speed $|v_i|$ across the width of the channel $S(x_j) \equiv \langle |v_i| \rangle_{i \in \text{bin } j}$ where $x_j$ is the horizontal $x$ coordinate of the $j$th bin and the $\langle |v_i| \rangle_{i \in \text{bin } j}$ represents the average speed of tracers in the $j$th bin. Then we normalize the speed profile by rescaling the speed profile to be 1 in the active zone and 0 in the inactive zone: $S^*(x) \equiv [S(x) - s_{\text{in}}]/[s_a - s_{\text{in}}]$, where $s_a$ is the average of speed profiles in the active zone and $s_{\text{in}}$ is the average of speed profiles in the inactive zone (Fig. 1e). Then we define the width of the active-inactive fluid interface as where the normalized speed profile is between 0.2 and 0.8 (dashed lines in Fig. 1e inset) and the position of the interface $x_1$ as where the normalized speed profile is 0.5 (solid line). The interface position is then analyzed for each frame, which allows us to determine the interface displacement $\Delta x \equiv x_1(\Delta t) - x_1(0)$ vs. time $t$ (Fig. 2a inset) and the squared interface displacement $\Delta x^2$ vs. time $t$ (Fig. 2a). To determine the interface progression exponent, $\gamma$, we fit $\log(\Delta x^2)$ vs. $\log(t)$ data to $\log(\Delta x^2) = \log(2P_1) + \gamma \log(t)$, with $P_1$ and $\gamma$ as fitting parameters (Figs. 2b and 4). To determine the interface progression coefficient, $P_1$, we assume $\gamma = 1$ and fit $\Delta x^2$ vs. $t$ data to $\Delta x^2 = 2P_1 t$ with $P_1$ as the fitting parameter (Figs. 2c, 3c, d).

## Generate flow speed map of active-inactive fluid system

To visualize the activity distribution in our active-inactive fluid system, we analyze the distribution of flow speed to generate flow speed maps (Fig. 1d). To complete this analysis, we analyze the flow velocities of fluids by analyzing the tracer motions in sequential images with a particle image velocimetry algorithm[70], which reveals the flow velocity field $\mathbf{V}(\mathbf{r},t)$ and associated distribution of flow speed $|\mathbf{V}(\mathbf{r},t)|$ in each frame. A heat color bar (Fig. 1d) is used to plot the speed distributions into color maps to reveal the evolution of speed distribution from the pre-activated state (black) to the homogeneously activated state (red/yellow).

## Numerically solve the Fick's law equations

We model diffusion-dominated active-inactive fluid mixing with the Fick's law equation, which requires us to solve for the concentration of ATP (Eq. (1)). To simplify the modeling, we consider a one-dimensional active fluid system confined in a segment $x = 0-L$ where $L = 20$ mm, the length of our experimental sample. Given that ATP is confined in the segment, we apply no-flux boundary conditions to the ATP concentrations (Eq. (2)). In the experiment, we expose the left side of the sample to UV light to release ATP, so in our model the ATP concentration has a step function as its initial condition (Eq. (3)). With the initial condition and boundary conditions, we solve the Fick's law equation to determine the spatial and temporal distribution of ATP. We used Mathematica 13.0 to numerically solve this differential equation with the NDSolveValue function by feeding Eqs. (1–3) into the function followed by specifying the spatial and temporal domains, which outputs the numerical solution of ATP concentration $C(x,t)$ and allows us to determine the evolution of ATP distribution (Fig. 3a). Then we convert the ATP distribution to mean speed distribution of active fluid by the Michaelis-Menten kinetics (Eq. (4); Fig. 3b inset), which shows a uniform mean speed in the left side of the sample followed by gradual activation of the right side of the system until a uniform state is reached (Fig. 3b). Then we define the position of the active-inactive fluid interface $\Delta x$ as where the mean speed decays to a half (Fig. 3c inset), which allows us to plot the squared interface displacement as a function of time (Fig. 3c). The plot in log-log axes exhibits a line with a unit slope, which suggests that the squared interface displacement is linearly proportional to time. By assuming this linear relation, we determine the interface progression coefficient $P_1$ by fitting $\Delta x^2 = 2P_1 t$ with $P_1$ as the fitting parameter. Finally, we repeat the calculation with different initial ATP concentrations in the left side of the system, $C_0$, and plot the interface progression coefficient, $P_1$, as a function of $C_0$. This plot allows us to compare the simulation results with the experimental measurements to examine the validity of our Fick's law-based model (Fig. 3d).

## Numerically solve the active nematohydrodynamic equations

To model the mixing of active and inactive fluids, we adopt Varghese et al.'s active fluid model to include the dynamics of the ATP concentration field[22]. Our model has four main components: (1) the kinetic equation describing the kinematics of self-elongating rods that flow and orient with the solvent as well as diffuse translationally and rotationally (Eqs. 6), (2) the Stokes equation describing how the solvent is driven by the active stress exerted by the self-elongating rods (Eqs. 7), (3) the relation between $\alpha$ and ATP that describes how the active stress depends on the nonuniform ATP distribution (Eqs. (8)), and (4) the

continuity equation of ATP transport that describes how ATP diffuses as well as flows with the solvent (Eq. (9)). We numerically solve these coupled equations with appropriate boundary and initial conditions using the finite element method by first converting them to their weak forms and then implementing them symbolically in COMSOL Multiphysics™ [60]. We show below the weak form of the convection-diffusion equation governing the evolution of ATP concentration field:

$$
\int_{\Gamma} dx^* dy^* \tilde{T} \frac{\partial C}{\partial t^*} = - \int_{\Gamma} dx^* dy^* D^* \left( \frac{\partial \tilde{T}}{\partial x^*} \frac{\partial C}{\partial x^*} + \frac{\partial \tilde{T}}{\partial y^*} \frac{\partial C}{\partial y^*} \right) \\
- \int_{\Gamma} dx^* dy^* \tilde{T} \left( u_x^* \frac{\partial C}{\partial x^*} + u_y^* \frac{\partial C}{\partial y^*} \right),
$$

(10)

where $\tilde{T}(x^*, y^*)$ is the test function and $\Gamma$ represents the system spatial domain. After solving these equations, we determine the spatial and temporal distributions of ATP concentrations and active fluid flow speeds (Fig. 6a and Supplementary Movie 4), which allows us to explore how the activity level of active fluid and molecular diffusion of ATP influences the mixing process of ATP in nonuniform active fluid systems (Fig. 6b, c).

### Simulate dispersion of checkerboard-patterned ATP

We apply our established hydrodynamic model to simulate how the initially checkerboard-patterned ATP disperses in active fluid and inactive fluid (Fig. 8). In this exploration, we use the same equations for **Q**, **u** and $C$ (Eqs. (6–9)), along with their initial and boundary conditions, except that $C$ is initialized in a checkerboard pattern in a 45 × 45 simulation box. We use two different checkerboard patterns for different trials. One has the $xy$-axis origin in the center of a grid square:

$$
C(x,y,t=0) = C_0 \bmod \left\{ \bmod \left[ \text{ceil} \left( \frac{x^*}{a^*} - 0.5 \right), 2 \right] + \bmod \left[ \text{ceil} \left( \frac{y^*}{a^*} - 0.5 \right), 2 \right] + 1, 2 \right\},
$$

(11)

where $C_0 = 5$ mM is the initial ATP concentration, $a^*$ represents the dimensionless grid size of the checkerboard pattern, mod($i,j$) represents the remainder of $i$ divided by $j$, and ceil($x$) represents the rounding of $x$ toward positive infinity (e.g., Fig. 8a with $a^* = 22$). The other checkerboard pattern has the $xy$-axis origin at a vertex of the grid:

$$
C(x,y,t=0) = C_0 \bmod \left\{ \bmod \left[ \text{ceil} \left( \frac{x^*}{a^*} \right), 2 \right] + \bmod \left[ \text{ceil} \left( \frac{y^*}{a^*} \right), 2 \right] + 1, 2 \right\}.
$$

(12)

The simulation is performed for the dimensionless grid size $a^*$ ranging from 2 to 22 for both active ($\alpha^* = 25$) and inactive ($\alpha^* = 0$) systems (Fig. 8a). We analyze the corresponding mixing time (averaged over 2 trials) as a function of $a^*$ (Fig. 8b) to reveal how the $a^*$-dependence of mixing times differs in active and inactive fluid systems.

### Data availability

The datasets generated and analyzed during the current study are available in the figshare repository, https://doi.org/10.6084/m9.figshare.21358017, or available from the corresponding author on request.

### Code availability

The Mathematica script used to solve the Fick's law-based model (Eqs. (1–4)) and the COMSOL file used to solve the active nematohydrodynamic equations coupled with active transport of ATP (Eqs. (6–9)) has been deposited on figshare (https://doi.org/10.6084/m9.figshare.20332806).

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

## Acknowledgements

T.E.B. and K.-T.W. would like to thank Drs. Seth Fraden and Aparna Baskaran of Brandeis University for insightful discussions on experiments and modeling in this manuscript. T.E.B. and K.-T.W. would also like to thank Dr. John Berezney of Brandeis University for assisting us in developing the UV light setup (Supplementary Fig. 10). We thank Ellie Lin from Lin Life Science for her assistance on editing the manuscript to enhance its flow and readability. T.E.B., E.H.T., and K.-T.W. acknowledge support from the National Science Foundation (NSF-CBET-2045621). This research is performed with computational resources supported by the Academic & Research Computing Group at Worcester Polytechnic Institute. We acknowledge the Brandeis Materials Research Science and Engineering Center (NSF-MRSEC-DMR-2011846) for use of the Biological Materials Facility. Parts of the work by C.-C. C. are funded and supported through the National Science and Technology Council (NSTC), Taiwan, under grant No. 111-2221-E-006-102-MY3, and through the 2022 Early Career Award from the College of Engineering and the Headquarters of University Advancement at National Cheng Kung University, which is sponsored by the Ministry of Education, Taiwan. M.M.N. is supported by the U.S. Department of Energy, Office of Science, Office of Basic Energy Sciences under Award No. DE-SC0022280.

## Author contributions

T.E.B., M.E.V., and K.-T.W. performed the research and designed the experiments; M.E.V. initiated the experiments; T.E.B., M.E.V., E.H.T. and J.H.D. collected experimental data; T.E.B., M.E.V., and K.-T.W. organized and analyzed the data; T.E.B., E.H.T., C.-C.C., M.M.N. and K.-T.W. established the continuum simulation platform on modeling nonuniform active fluid systems; T.E.B. and K.-T.W. wrote the manuscript; and K.-T.W. supervised the research. All authors reviewed the manuscript.

## Competing interests

The authors declare no competing interests.
