## [Peer Review File · Nature Communications]

Self-mixing in microtubule-kinesin active fluid from nonuniform to uniform distribution of activityREVIEWER COMMENTS

Reviewer #1 (Remarks to the Author):

In this paper, the authors perform experiments on mixing in an active fluid composed of microtubules and kinesin clusters powered by ATP. Using light activated ATP, they are able to turn on activity over a finite region of space, thus creating an interface between an active fluid and a passive fluid. As the light-activated ATP disperses across the interface due to both diffusion and active mixing, the region where active flows take place expands spatially, thereby further enhancing transport and mixing. The authors characterize the rate at which the ATP gets mixed between the two regions, and find that the mixing is diffusive at low levels of activity, but superdiffusive at high levels of activity. They compare their experimental results with two types of models: the solution of a simple 1D diffusion equation, and a more complex PDE model for an active nematic coupled to fluid flow.

Overall, the experiments are impressive and well executed, and the paper is well written. The main result, namely the transition from diffusive to superdiffusive spreading with increasing of activity, is interesting, although perhaps not entirely surprising. However, the discussion and analysis as well as the comparison with models are sometimes a bit weak, and it is not entirely clear what fundamental new physics is learned as a result of this study. In particular, the transition to superdiffusive transport is not really understood (and in fact it is not captured by the models). As a result, I cannot recommend publication of the manuscript in its present form in Nature Communications.

Here are some more detailed comments and suggestions that the authors should addressing to improve the quality of the paper:

1. It's not completely clear how to interpret the velocity of equation (4): ATP should directly affect the velocity of kinesin motors, but how it translates into the velocity the interface is not trivial: indeed, the motors produce local microtubule motions, which then interact through the fluid the causing interfacial spreading. Furthermore, the fact that the interface in the 1D model spreads diffusively is not very surprising, since the model is based on the diffusion equation and the velocity of equation (4) is directly tied to the ATP concentration (in fact, when $[ATP] \ll k$ in the early stages of mixing, the velocity is directly proportional to $[ATP]$). As a result, I'm not sure exactly what is learned from the model.
2. It is likely that the transition to superdiffusion is tied to the emergence of strong large-scale motions in the fluid, which result in advective (ballistic) transport on top of the diffusive spreading. This is supported by the observation on page 4 that in order to increase the velocities in the fluid the authors increasing the thickness of the sample. This should have a direct effect on the correlation length scales of the turbulent motions in the active fluid (this point is very briefly addressed in the discussion section). It would be very interesting to characterize these length scales and determine how they affect the mixing process.
3. The fact the checkerboard patterns result in faster mixing than a single interface is again not very surprising, since gradients are introduced on short length scales.
4. The discussion of rheology in the discussion section is quite interesting, in particular the fact that the microtubule network really behaves as crosslinked gel until the kinesin motors are activated. However, it is not completely obvious why this would contribute to superdiffusive transport as hypothesized by the authors: in fact, I would expect transport to

be weaker due to crosslinking.

Reviewer #2 (Remarks to the Author):

The manuscript reports the results of a combined experimental and theoretical/numerical study of fluid mixing by active fluids. Ultraviolet light-activated caged ATP and fluorescent dyes are used to activate regions of a microtubule-kinesin active fluid. Both a system with an interface between an activated and a passive microtubule-kinesin mixture, and a checkerboard activation pattern are investigated. As activity is turned on, the interfaces broaden and move into the initially inactivated regions. At low activation levels, mixing is found to be governed by active diffusion-like processes at the active-inactive interface, while at higher activity levels, superdiffusion-like processes dominate.

Samples activated in a checkerboard pattern reach homogeneity faster than those with a single dividing interface.

A model of active nematohydrodynamics coupled to ATP transport is employed to describe the coupled mixing process numerically. The results are in qualitative agreement, but differ on the quantitative level significantly.

Mixing of fluids on the microscale is difficult to achieve, because Reynolds number is low and mixing is dominated by molecular diffusion. Active fluids have the potential to generate a pronounced speed-up of the mixing process. The current study focuses on systems with spatiotemporally varying activities and the role of interfaces in the mixing process. This is a very interesting aspect, which has not received much attention so far.

The following questions and comments should be addressed:

- (1) Why would a diffusive MSD of the interface position, which would naively be expected for an underlying diffusive process, give any hint about a super-diffusive process?
- (2) How does the flow speed, which is determined by Michaelis-Menten kinetics, enter into model given by eqs. (1-3)?
- (3) What changes at the molecular scale at the velocity threshold of 5 micron/sec?
- (4) Why does mixing time increase linearly with grid size (Fig. 7b), and extrapolate to zero mixing time at a grid size of about 0.9 mm ?
- (5) The theoretical model predicts a scaling of interface progression coefficient, which is consistent with the results from experiments. However, the coefficient magnitudes differed by a factor of 5 between the model and the experiments.

Furthermore, the simulation does not show the transition from an active diffusion-like to a superdiffusion-like processes observed experimentally.

The authors propose as a possible explanation that the rheology of microtubule network is not taken

into account in the numerical model. Is this a plausible explanation, in particular for the factor 5?

(6) The authors emphasize that the mixing efficacy of the nonuniform active fluid systems depends on the distribution of activity, with systems consisting of more small active areas

evolve to a homogeneous state faster than systems with the same total active area distributed as

one piece. They conclude that the activity-uniform active fluid has the highest mixing performance.

However, this is exactly the behavior I would expect for a passive systems: As interfaces propagate diffusively, shorter distances can be covered more quickly. Thus, what is the significance

of the active process in this general conclusion?

(7) Arguments about maximizing system entropy are difficult because this a non-equilibrium active system.

(8) The following references might also be relevant:

-- A. Doostmohammadi et al., Nat. Commun. 8, 15326 (2017);

-- G.A. Vliegenthart et al., Sci. Adv. 6, eaaw9975 (2020);

-- H. Reinken et al., Commun. Phys. 3, 76 (2020);

-- R. Alert et al., Nature Physics 16, 682 (2020);

-- K. Qi et al., Commun. Phys. 5, 49 (2022).

Reviewer #3 (Remarks to the Author):

The authors have performed mixing experiments on microtubule-kinesin based active nematics by taking advantage of an interesting molecule called “caged ATP”. This ATP can be introduced uniformly into the system in its inactive state and then activated remotely via a UV light to give spatial patterning of activity in the active nematic system.

They observed diffusion-like and super-diffusive regimes dependent on the speed of the microtubule flows and compare results with some simulations of the system with a variety of quantitative experiments. Beautiful videos demonstrate the concepts very well. The data are fit to a fairly simple transport model that seems to describe the behaviors well. An additional and interesting experiment towards the end of the paper tests checker-board patterns of activation to look at how fast the system reaches homogeneity as a function of active/inactive interfacial length. This section of the paper could probably be a separate publication if explored in more detail.

Overall I think that the paper covers an exciting topic and one that should be focused on by the active nematic community. Self-mixing in active nematics represents an important direction in the evolution of this field. I like the hypothesis driven approach and the data is presented in a logical and interesting fashion. A large amount of data is shown with a focus largely on statistical measures such as 1D ‘diffusion’ constants (i.e. P in this case).

I found a few issues in the paper that should be corrected pre-publication but I think that the paper is appropriate for Nature Communications in scope and impact.

1. The title of the paper can be improved for clarity - is it missing a comma or colon? I would use "activity" not "activities"

2. I found that the description of the results at the start of that section to be a bit lacking in necessary detail. The authors need to be clearer about the nature of the "initially inactivated fluid" in the first results paragraph. Is it mixed Mts and kinesin but not formed into bundles (random filaments, or small bundles not yet aligned)? Presumably the solution has not yet been in the active nematic state? Are the kinesin clusters bound to MTs yet but the Mts are isotropic? It appears from the video that this is the case and I can guess, but clarification must be added. It seems that initiation of activity is not a reversible process - i.e. once bundled the system will never go back to the initial state (e.g. when ATP runs out). This can also be clarified.

3. In the first results paragraph we also need to know where the dyes are located to interpret the videos and Figures easily - is it on the MT? The kinesin or perhaps in the water. This needs to be added in the first results paragraph. How about the tracer particles? Are they coupled to the MTs? How are they confined to the active layer? I realize that there is plenty of detail in the methods section at the end of the paper but the most important points for figure interpretation should be in the main text otherwise it's too difficult to read the paper - especially for the uninitiated general reader.

4. The authors don't address the concept of chaotic mixing and advective flows at all in the paper and I found that to be an omission - how does that concept relate to the measures for quantifying mixing in the submitted work? A short discussion might help put this work in context with that recent work cited as ref 24 where chaotic mixing was described for the same system. Some other groups are also considering advective flows in these systems. How would this be related to the present work? I also think that the idea that mixing is driven primarily by defects should be addressed. Can this idea be related to the present work? This was first introduced in theoretical works by Marchetti and shown experimentally in ref 24. Since the submitted paper is not the first to talk about self-mixing in active nematics, a discussion should be added to see how these different papers fit together and can be connected to the submitted work.

5. The results presented in the paper focus on large length-scales (i.e. much larger than the active length scale. Can the authors discuss their results where the length scales approach the active length scale or even go below it? Transport measures may be different on small scales and should be ballistic-like. Can local flows on the scale of defects be tracked to get more detail on these scales?

6. It appears that Fig 3d might fit to Michaelis-Menten kinetics, it looks like the trend is approximately there. Is that trend expected for this system? Comment on the shape of the curve and prior work.

7. In all the Figures the captions need to make it much clearer which panels are calculations and which are experimental data. The reader should not have search around for this information.

Line 241 - You can't use "turbulences" - this should be phrased better - do you mean vortices? Areas of "active turbulence". Please clarify.

Fig S3 - the plural of spectrum is spectra.

Reviewer 1:

In this paper, the authors perform experiments on mixing in an active fluid composed of microtubules and kinesin clusters powered by ATP. Using light activated ATP, they are able to turn on activity over a finite region of space, thus creating an interface between an active fluid and a passive fluid. As the light-activated ATP disperses across the interface due to both diffusion and active mixing, the region where active flows take place expands spatially, thereby further enhancing transport and mixing. The authors characterize the rate at which the ATP gets mixed between the two regions, and find that the mixing is diffusive at low levels of activity, but superdiffusive at high levels of activity. They compare their experimental results with two types of models: the solution of a simple 1D diffusion equation, and a more complex PDE model for an active nematic coupled to fluid flow.

Overall, the experiments are impressive and well executed, and the paper is well written. The main result, namely the transition from diffusive to superdiffusive spreading with increasing of activity, is interesting, although perhaps not entirely surprising. However, the discussion and analysis as well as the comparison with models are sometimes a bit weak, and it is not entirely clear what fundamental new physics is learned as a result of this study. In particular, the transition to superdiffusive transport is not really understood (and in fact it is not captured by the models). As a result, I cannot recommend publication of the manuscript in its present form in Nature Communications.

We thank the reviewer for their assessment. The essence of our work was to investigate the mixing dynamics of active fluid systems with a dynamic activity gradient. Recently, there has been a trend within active fluid research to study systems with activity gradients. For example, Zhang *et al.* Nature Materials 20, 875 (2021) and Ross *et al.* Nature 572, 224 (2019) investigated the role of activity gradients in the self-organization of active fluid and Shankar and Marchetti PRX 9, 041047 (2019) predicted the defect dynamics in an activity gradient. However, most existing work about activity gradients is focused on systems with steady state, time-independent activity gradients. Our work is novel in that it explores an activity gradient that changes spontaneously with time. Also, this paper hints that melting dynamics of the microtubule network play a role in the behavior of a dynamic activity gradient, and that such dynamics have been overlooked in existing active fluid models. As such, we believe our work can promote development of active fluid modeling to include the network melting mechanism and thus more accurately describe complex active fluid systems with dynamic distributions of activity.

Here are some more detailed comments and suggestions that the authors should addressing to improve the quality of the paper:

1. It's not completely clear how to interpret the velocity of equation (4): ATP should directly affect the velocity of kinesin motors, but how it translates into the velocity the interface is not trivial: indeed, the motors produce local microtubule motions, which then interact through the fluid the causing interfacial spreading.

We thank the reviewer for the comment. The velocity in Eq. 4 represents mean speed of active fluid flow driven by extensile microtubules whose motions are driven by kinesin motor proteins consuming local ATP and stepping along microtubules (Fig. 1a). The progression of the interface is a complex phenomenon that involves the coupling of ATP dispersion and active fluid flows. We agree with the reviewer that how dispersion of ATP relates to progression of the interface is not trivial. To further explore the issue, we derived an analytical expression for interface progression coefficient, P_I , (Eq. S7 in Supplementary Discussion 3) that showed that P_I depended on the initial ATP concentration C_0 via an inverse complementary error function. Then we explored how P_I varied with C_0 numerically, which revealed that

a higher initial ATP concentration led to a faster-progressing interface (Fig. 3d). Surprisingly, we further found that the interface progression coefficients P_I are well fit to the Michaelis-Menten equation (the second curve from the top in Supplementary Fig. 4a). This result demonstrated that how the speed of active fluid depended on ATP concentration directly led to how the interface progression depended on ATP concentration. For readers who are interested in the connection between these two ATP dependences, we added two supplementary discussions (Supplementary Discussions 2 & 3) to explore the connection numerically as well as analytically.

Furthermore, the fact that the interface in the 1D model spreads diffusively is not very surprising, since the model is based on the diffusion equation and the velocity of equation (4) is directly tied to the ATP concentration (in fact, when $[ATP] \ll k$ in the early stages of mixing, the velocity is directly proportional to $[ATP]$). As a result, I'm not sure exactly what is learned from the model.

We thank the reviewer for asking about the significance of our 1D Fick's law-based active fluid model. The purpose of introducing this model was to demonstrate that the mixing of active and inactive fluids we observed (Fig. 2) could be described with a minimal model considering only diffusion of ATP and Michaelis-Menten kinetics (Fig. 3), without the need for a more complex hydrodynamic model such as the one we developed later in the manuscript (Fig. 6). We found that the simple 1D Fick's law-based active fluid model did agree with experimental observations (Fig. 3d). This consistency between the Fick's law-based model and our experimental results suggests that Michaelis-Menten kinetics successfully approximate the conversion of ATP concentration into local active fluid speed. For readers who are interested in learning about why we introduced the Fick's law-based model, we revised the manuscript to clarify the motivation of developing the model and its associated intellectual merits in the main text (lines 100-104 and 129-133) as well as in Supplementary Discussion 2.

In response to the reviewer's comment that the connection between diffusion-like interface progression ($\gamma = 1$) and diffusive ATP ($a = 1$) is an expected, trivial consequence of adopting the Michaelis-Menten equation (Eq. 4), we performed additional studies (Supplementary Discussion 2) that showed that the connection was not necessarily built upon using the Michaelis-Menten equation, because even when the relation between ATP concentration and flow speed became square root, quadratic, cubic, or higher power-law dependent, the resulting interface progression exponent remained $\gamma = 1$ (Supplementary Fig. 3d inset). These additional studies show that connection between $\gamma = 1$ and $a = 1$ was not sensitive to which flow speed-ATP relation we chose (although the interface progression coefficient P_I was highly sensitive to the relation [Supplementary Fig. 3d]). For readers who are interested in learning about the role of flow speed-ATP relation in our model, we added Supplementary Discussion 2 to demonstrate how the modeling results would be influenced with 10 other relations. We thank the reviewer for questioning the intellectual merit of our Fick's law-based model which drove us to enrich the exploration of our model and strengthen the manuscript.

2. It is likely that the transition to superdiffusion is tied to the emergence of strong large-scale motions in the fluid, which result in advective (ballistic) transport on top of the diffusive spreading. This is supported by the observation on page 4 that in order to increase the velocities in the fluid the authors increasing the thickness of the sample. This should have a direct effect on the correlation length scales of the turbulent motions in the active fluid (this point is very briefly addressed in the discussion section). It would be very interesting to characterize these length scales and determine how they affect the mixing process.

We greatly thank the reviewer for the insightful comment. The reviewer was correct that increasing the sample container height would affect the correlation length scales of the turbulent motion. We added a supplementary discussion (Supplementary Discussion 4) in which we quantify how the correlation length

increased with sample container height (Supplementary Fig. 5d). To explore how the increased correlation length would affect the mixing of active-inactive fluid systems, we adopted a dimensionless quantity, the Péclet number (Pe), defined as $Pe \equiv \bar{v}_{ab}l_c/D$, where \bar{v}_{ab} represents the mean flow speed of active fluid, D is the diffusion coefficient of ATP, and l_c is the correlation length of flow velocity. The physical interpretation of this quantity is the ratio of convective transport rate to diffusive transport rate. When the Péclet number is greater than of order 1, the active transport is dominated by convection, whereas when the Péclet number is smaller than of order 1, the active transport is dominated by diffusion. To explore how the behaviors of active transport affect mixing in an active-inactive fluid system, we first analyzed the interface progression exponent γ as a function of Péclet number (Fig. 4b) and found that as $Pe \lesssim 3$, $\gamma \approx 1$ which corresponded to the diffusion-like mixing captured in our Fick's law-based model (Fig. 3). Interestingly, as the Péclet number is increased to $Pe \gtrsim 3$, we observed γ became greater than 1, which corresponded to the transition in interface progression from diffusion-like to superdiffusion-like. As such, the reviewer was correct; the transition to superdiffusion was tied to the gradual emerging convection mechanism that governed the mixing process of active-inactive fluid system. We have revised the manuscript to clarify the underlying mechanism causing the transition (lines 146-161 with accompanying Fig. 4b). We greatly appreciate the reviewer's constructive feedback that significantly improved the manuscript.

3. The fact the checkerboard patterns result in faster mixing than a single interface is again not very surprising, since gradients are introduced on short length scales.

We agree with the reviewer that the results of the checkerboard pattern experiments were expected and thus the intellectual contribution was minimal. The reason we introduced these experiments was to explore how the distribution of activity would affect the mixing process of the active-inactive fluid system, as previously we had only explored one configuration of activity distribution (one side active and one side inactive).

To enrich this part of the work, we further explored the checkerboard system in our established simulation platform (Fig. 8), revealing two interesting results: (1) At a sufficiently small grid size, the simulated mixing times of active and inactive systems were indistinguishable. We believe that this is because the active fluid needs time to "warm up". In experiments, the system had a warm-up time caused by network melting (Supplementary Discussion 6). Although a network melting mechanism was not included in the model, the simulated active fluid flow still took dimensionless time to rise because the onset of the flows was triggered by the initial activity-driven instability in extensile Q field which took finite dimensionless time to develop (~ 1 dimensionless time in this case; Supplementary Video 6), and during the warm-up time, molecular diffusion is the main driving force of mixing. When the grid size was very small, the mixing could be completed by diffusion before the fluid activity could speed up the mixing. Thus, the simulation based on the checkerboard pattern suggested a limitation of active fluid mixing: The mixing effect of active fluid can only take place when the mixture is sufficiently nonuniform. (2) The checkerboard-pattern simulation also revealed that the mixing process of active fluid is less susceptible to changes in grid size than in inactive fluid systems (Fig. 8b). Altering the dimensionless grid size from 2 to 22 changed the mixing time of the inactive fluid system by a factor of 40 but only changed the mixing time of the active fluid system by a factor of 3. This result suggests that an active fluid system is less susceptible to the initial distribution of the mixture than an inactive fluid system.

These two new results (added to Results lines 261-268, Discussion lines 307-319, Methods lines 516-529) demonstrate limits and advantages of introducing active fluid to microfluidic mixing systems with nonuniform activity. We thank the reviewer for the comment that drove us to enrich the manuscript. We believe that the checkerboard work is now more relevant and could inspire further exploration of the active mixing process with other complex distributions of activity.

4. The discussion of rheology in the discussion section is quite interesting, in particular the fact that the microtubule network really behaves as crosslinked gel until the kinesin motors are activated. However, it is not completely obvious why this would contribute to superdiffusive transport as hypothesized by the authors: in fact, I would expect transport to be weaker due to crosslinking.

We thank the reviewer for asking this question. After following the reviewer's second comment, we found that the transition to superdiffusion-like interface progression ($\gamma > 1$) was tied to the underlying transition of active transport from being diffusion-like ($Pe \lesssim 3$) to superdiffusion-like ($Pe \gtrsim 3$), and thus the network rheology might not have played a significant role in this transition. Thus, we removed that hypothesis from the manuscript. Nevertheless, we explored our data more deeply and found that the network rheology could indeed slow the progression of active-inactive interface (Supplementary Fig. 7e), as the reviewer expected, and this slower progression was not captured in our active-fluid hydrodynamic model because the model did not include the network melting mechanism (Supplementary Fig. 8c). These results demonstrate that the network melting mechanism plays a role in the mixing process of activity-nonuniform active fluid systems, and this mechanism has thus far been overlooked in the modeling of microtubule-kinesin active fluid systems. In the revised manuscript, we addressed the significance of the network rheology in mixing dynamics (lines 284-297 and Supplementary Discussion 6) and suggested potential future experiments and simulations to further investigate the network melting dynamics (lines 297-302). We believe that our current manuscript will encourage the active fluid community to investigate this topic further and inspire further exploration in self-organization of activity-nonuniform active fluid systems.

Reviewer 2:

The manuscript reports the results of a combined experimental and theoretical/numerical study of fluid mixing by active fluids. Ultraviolet light-activated caged ATP and fluorescent dyes are used to activate regions of a microtubule-kinesin active fluid. Both a system with an interface between an activated and a passive microtubule-kinesin mixture, and a checkerboard activation pattern are investigated. As activity is turned on, the interfaces broaden and move into the initially inactivated regions. At low activation levels, mixing is found to be governed by an active diffusion-like process at the active-inactive interface, while at higher activity levels, superdiffusion-like processes dominate. Samples activated in a checkerboard pattern reach homogeneity faster than those with a single dividing interface. A model of active nematohydrodynamics coupled to ATP transport is employed to describe the coupled mixing process numerically. The results are in qualitative agreement, but differ on the quantitative level significantly.

Mixing of fluids on the microscale is difficult to achieve, because Reynolds number is low and mixing is dominated by molecular diffusion. Active fluids have the potential to generate a pronounced speed-up of the mixing process. The current study focuses on systems with spatiotemporally varying activities and the role of interfaces in the mixing process. This is a very interesting aspect, which has not received much attention so far.

We thank the reviewer for their comments on the intellectual merit of our work. Please find our point-by-point responses to the reviewer's feedback below.

The following questions and comments should be addressed:

(1) Why would a diffusive MSD of the interface position, which would naively be expected for an underlying diffusive process, give any hint about a super-diffusive process?

We thank the reviewer for their comment. We have revised the paper to clarify this point. In the previous version of the paper, we used a proof by contradiction: proving a point by demonstrating that its opposite is a contradiction. We sought to prove that $\gamma = 1$ is the consequence of ATP diffusion ($a = 1$) by first stating that $\gamma = 1$ because $a > 1$ (superdiffusion). Then we used the Fick's law model to show that this statement was incorrect; thus, the invalidity of the statement indicated that $\gamma = 1$ from $a = 1$ (or $a \leq 1$, more strictly).

We agree with the reviewer that this proof was too complicated and led to unnecessary confusion, and we removed the proof from the manuscript. As an alternative, we revised the Fick's law model section in the manuscript to directly show that modeling ATP dispersion with the diffusion equation leads to diffusion-like interface progression (lines 100-133). To provide deeper insight into the connection between ATP dispersion and interface progression, we presented an algebraic proof that squared interface displacement is proportional to time: $\Delta x^2 \propto t$ (Eq. S6) and provided an analytical perspective on how diffusive ATP transport led to diffusion-like interface progression (Supplementary Discussion 3). We believe that the manuscript is now clearer and more understandable.

(2) How does the flow speed, which is determined by Michaelis-Menten kinetics, enter into model given by eqs. (1-3)?

We thank the reviewer for asking. The Michaelis-Menten kinetics equation is separate from Eqs. 1-3. The reason we modeled the ATP-dependent flow speed with Michaelis-Menten kinetics is because our previous studies of the flow speed of active fluid for various ATP concentrations (Figs. 3A-C in Bate *et al.* Soft Matter 15, 5006 [2019]) showed that the flow speed of active fluid followed Michaelis-Menten kinetics for ATP concentrations greater than 100 μM . Additionally, we believe this choice of model is reasonable

because the active fluid is driven by the motion of microtubules, which are driven by kinesin motors whose stepping rates are known to follow the Michaelis-Menten kinetics (Schnitzer *et al.* Nature 388, 386 [1997]), and thus these kinetics could be scaled up to active fluid scale.

To clarify how our choice of flow speed-ATP relation affected the modeling results in terms of predicted interface progression exponents (γ) and coefficients (P_I), in the revised manuscript we explored ten other plausible relations to connect ATP and flow speed and found that the Michaelis-Menten equation led to the best match between the predictions and the experimental results (Supplementary Discussion 2, Supplementary Fig. 3d). We are now more confident that the Michaelis-Menten equation is an appropriate coarse-grained approach to model local flow speed in terms of ATP concentrations. We thank the reviewer for pointing out the weakness in our manuscript which drove us to make the manuscript more robust.

(3) What changes at the molecular scale at the velocity threshold of 5 micron/sec?

We thank the reviewer for asking this important question. In the previous version of manuscript, we stated that the active-inactive interface progression occurred at an active fluid flow speed of 5 $\mu\text{m/s}$. In the revised version of the manuscript, we adopted a new perspective on this issue: we found that the transition is in fact related to Péclet number rather than active fluid speed. Péclet number is defined as $Pe \equiv \bar{v}_{ab}l_c/D$, where \bar{v}_{ab} represents the mean flow speed of active fluid, D is the diffusion coefficient of ATP, and l_c is the correlation length of flow velocity. The physical interpretation of this quantity is the ratio of convective transport rate to diffusive transport rate. When the Péclet number is smaller than of order 1, the active transport is dominated by diffusion, whereas when the Péclet number is greater than of order 1, the active transport is dominated by convection. To explore how the behaviors of active transport affect mixing in an active-inactive fluid system, we first analyzed the interface progression exponent γ as a function of Péclet number (Fig. 4b) and found that as $Pe \lesssim 3$, $\gamma \approx 1$, which corresponds to the diffusion-like mixing captured in our Fick's law-based model (Fig. 3). Interestingly, as the Péclet number increased to $Pe \gtrsim 3$, γ became greater than 1, which corresponded to the transition in interface progression from being diffusion-like to being superdiffusion-like. Our further analysis revealed that the transition from diffusion-like to superdiffusion-like progression of the active-inactive interface was not directly relevant to the 5 $\mu\text{m/s}$ flow speed. Instead, it was due to the transition of active transport from diffusion-dominated to superdiffusion-dominated, as characterized by a dimensionless Péclet number increasing from of order 1 to of order greater than 1. We revised the manuscript to clarify the mechanism underlying the diffusion-superdiffusion transition of the active-inactive interface (lines 146-161, Fig. 4b). We thank the reviewer for their question, which drove us to dive more deeply into our data, pursue a fuller understanding of interface progression, and make the manuscript more robust.

(4) Why does mixing time increase linearly with grid size (Fig. 7b), and extrapolate to zero mixing time at a grid size of about 0.9 mm ?

We thank the reviewer for the observations on our experimental checkerboard data (Fig. 7b). However, we think it was not clear that the mixing time varied linearly with grid size. We believe that as the grid size approached zero, the mixing time would vary more slowly and approach zero (see the figure below). To learn more about grid size-dependence of mixing time, we performed a corresponding checkerboard active fluid simulation (Fig. 8) and found that as the grid sizes decreased, the mixing time in active fluid system became indistinguishable with the mixing time in inactive fluid system where the mixing was driven only by molecular diffusion (Fig. 8b). We believe that this was because our active fluid system needed time to “warm up”. In experiments, the system had a warm-up time caused by network melting (Supplementary Discussion 6). Although a network melting mechanism was not included in the model, the simulated active fluid flow still took dimensionless time to rise because the onset of the flows was triggered by the initial

activity-driven instability in extensile Q field which took finite dimensionless time to develop (~ 1 dimensionless time in this case; Supplementary Video 6), and during the warm-up time, molecular diffusion is the main driving force of mixing. Thus, the simulation showed that as the grid sizes approached zero, the mixing dynamics would also change from active flow-dominated to molecular diffusion-dominated, so the mixing time was not expected to vary linearly with all grid sizes. We thank the reviewer for the comment on our checkerboard data which motivated us to explore the checkerboard system more deeply. For readers who are interested in learning about checkerboard mixing dynamics in the limit of small grid sizes, we added our simulation checkerboard work in Results (lines 261-268), Discussion (lines 307-319), and Methods (lines 516-529) along with accompanying figure (Fig. 8). We believe that our checkerboard work is now richer and can provide readers with deeper understanding of the mixing dynamics of nonuniform active fluid systems.

(5) The theoretical model predicts a scaling of interface progression coefficient, which is consistent with the results from experiments. However, the coefficient magnitudes differed by a factor of 5 between the model and the experiments. Furthermore, the simulation does not show the transition from an active diffusion-like to a superdiffusion-like processes observed experimentally. The authors propose as a possible explanation that the rheology of microtubule network is not taken into account in the numerical model. Is this a plausible explanation, in particular for the factor 5?

We thank the reviewer for raising the concern about the five-factor discrepancy of interface progression coefficient P_I between the model and experiment. In response to this comment, we examined our data more closely and concluded that the discrepancy mainly resulted from using an incorrect ATP diffusion coefficient. In the previous version of this manuscript, the model used the diffusion coefficient of ATP in water, because the base of the microtubule-kinesin active fluid is water. However, Gagnon *et al.* showed that the microtubule network has a viscosity greater than water (PRL 125, 178003, [2020]), which implies that the actual diffusion coefficient of ATP should be lower than its value in water. To better estimate the diffusion coefficient of ATP in our microtubule network, we performed extensive additional studies to explore the diffusion coefficient in inactive microtubule-kinesin fluid. We were not able to track ATP directly, so as an alternative we tracked fluorescein, which we could visualize with fluorescent microscopy. We found that the diffusion coefficient of fluorescein in the inactive microtubule network was one-fifth the value reported for aqueous solution (Supplementary Discussion 1), which suggested that the diffusion coefficient of ATP in our active fluid is also 5 times lower than the value reported in pure water. After making this correction, we found that our model was consistent with experimental observation (with $\sim 10\%$ relative difference; Fig. 3d). We revised the Fick's law model section in the manuscript (lines 100-133) and added a description of how we estimated the diffusion coefficient of ATP in our active fluid system

(Supplementary Discussion 1). We thank the reviewer for questioning about the weakness in our model which drove us to strengthen our model and make our finding more robust.

(6) The authors emphasize that the mixing efficacy of the nonuniform active fluid systems depends on the distribution of activity, with systems consisting of more small active areas evolve to a homogeneous state faster than systems with the same total active area distributed as one piece. They conclude that the activity-uniform active fluid has the highest mixing performance. However, this is exactly the behavior I would expect for a passive systems: As interfaces propagate diffusively, shorter distances can be covered more quickly. Thus, what is the significance of the active process in this general conclusion?

We thank the reviewer for questioning the significance of fluid activity in the checkerboard work. We performed the checkerboard experiments because throughout our investigation on active-inactive fluid system, we only focused on one configuration of activity distribution: One side activated and the other side inactive. Thus, we were curious about how the mixing process would have been different if the activity had been distributed in a different arrangement. In response to the reviewer's comment, we further explored the checkerboard mixing system via our active fluid simulation (Fig. 8) and compared the mixing of the active and inactive checkerboard-pattern systems, to delve more deeply into this topic and provide more relevant findings in the manuscript. Surprisingly, our simulation showed that the mixing time of the active fluid system depended less on the grid size than the mixing time of an inactive fluid system (Fig. 8b). Increasing the dimensionless grid size from 2 to 22 increased the mixing time of inactive fluid system by a factor of 40, whereas in the active fluid system, the mixing time was only increased by a factor of 3. Such drastic difference showed that the mixing efficacy of active fluid is less sensitive to changes in the initial condition of the mixture. We added these new checkerboard simulation results to Results (lines 261-268), Discussion (lines 307-319), and Methods (lines 516-529) along with accompanied figure (Fig. 8). We thank the reviewer for questioning about the intellectual merits of our checkerboard work which drove us to delve into the checkerboard mixing systems and enriched the checkerboard work in our manuscript.

(7) Arguments about maximizing system entropy are difficult because this a non-equilibrium active system.

We thank the reviewer for the correction. We removed arguments about entropy.

(8) The following references might also be relevant:

- A. Doostmohammadi et al., Nat. Commun. 8, 15326 (2017);
- G.A. Vliegthart et al., Sci. Adv. 6, eaaw9975 (2020);
- H. Reinken et al., Commun. Phys. 3, 76 (2020);
- R. Alert et al., Nature Physics 16, 682 (2020);
- K. Qi et al., Commun. Phys. 5, 49 (2022).

We thank the reviewer for bringing our attention to these important works. We have cited these works in relevant places in our manuscript.

Reviewer 3:

The authors have performed mixing experiments on microtubule-kinesin based active nematics by taking advantage of an interesting molecule called “caged ATP”. This ATP can be introduced uniformly into the system in its inactive state and then activated remotely via a UV light to give spatial patterning of activity in the active nematic system. They observed diffusion-like and super-diffusive regimes dependent on the speed of the microtubule flows and compare results with some simulations of the system with a variety of quantitative experiments. Beautiful videos demonstrate the concepts very well. The data are fit to a fairly simple transport model that seems to describe the behaviors well. An additional and interesting experiment towards the end of the paper tests checker-board patterns of activation to look at how fast the system reaches homogeneity as a function of active/inactive interfacial length. This section of the paper could probably be a separate publication if explored in more detail.

Overall I think that the paper covers an exciting topic and one that should be focused on by the active nematic community. Self-mixing in active nematics represents an important direction in the evolution of this field. I like the hypothesis driven approach and the data is presented in a logical and interesting fashion. A large amount of data is shown with a focus largely on statistical measures such as 1D ‘diffusion’ constants (i.e. P in this case).

I found a few issues in the paper that should be corrected pre-publication but I think that the paper is appropriate for Nature Communications in scope and impact.

We thank the reviewer for the support.

1. The title of the paper can be improved for clarity - is it missing a comma or colon? I would use "activity" not "activities"

We thank for the reviewer’s correction. We have changed the title to “Self-mixing in microtubule-kinesin active fluid from nonuniform to uniform distribution of activity.”

2. I found that the description of the results at the start of that section to be a bit lacking in necessary detail. The authors need to be clearer about the nature of the “initially inactivated fluid” in the first results paragraph. Is it mixed Mts and kinesin but not formed into bundles (random filaments, or small bundles not yet aligned)? Presumably the solution has not yet been in the active nematic state? Are the kinesin clusters bound to MTs yet but the Mts are isotropic? It appears from the video that this is the case and I can guess, but clarification must be added. It seems that initiation of activity is not a reversible process - i.e. once bundled the system will never go back to the initial state (e.g. when ATP runs out). This can also be clarified.

We thank the reviewer for highlighting the need to clarify the initial, inactive state of our active fluid system. Our inactive fluid contained microtubules that spontaneously formed bundles by depletion; these bundles were further crosslinked by kinesin motor dimers, forming an elastic gel network. We prepared the inactive gel sample in a test tube where microtubules were expected to orient isotropically, but when we loaded the mixture into the flow cell, shear flow was induced and drove the microtubules to align along the flow cell, so microtubules initially had a preferred alignment along the long edge of the flow cell (Supplementary Fig. 1). After the fluid was activated by ultraviolet light, the microtubule network underwent an irreversible process of becoming a 3D self-rearranging isotropic active gel consisting of extensile microtubule bundles that buckled and annealed repeatedly until the ATP ran out. We have clarified the initial state of microtubule network before UV activation along with various details about the network dynamics such as bundle formation and the irreversibility of the network structure in the main text (lines 55-77) and added a supplementary figure to show the microtubule alignment right after loading (Supplementary Fig. 1). We

noticed that Najma *et al.* recently posted an arXiv reporting the similar initial network structure (Fig. 1c in Najma *et al.* arXiv: 2112.11364 [2022]) so we also cited the arXiv for readers who wanted to know more details about the initial state of microtubule-kinesin active fluid. We believe that now the readers have access to sufficient information about the initial state of our inactive fluid. We thank the reviewer for the request that increased the clarity of our manuscript.

3. In the first results paragraph we also need to know where the dyes are located to interpret the videos and Figures easily - is it on the MT? The kinesin or perhaps in the water. This needs to be added in the first results paragraph. How about the tracer particles? Are they coupled to the MTs? How are they confined to the active layer? I realize that there is plenty of detail in the methods section at the end of the paper but the most important points for figure interpretation should be in the main text otherwise it's too difficult to read the paper - especially for the uninitiated general reader.

We thank the reviewer for addressing the issue of important methods details in the main text. We added explicit description in the main text that the microtubules were labeled with Alexa 647 (Line 69-70) and that tracers were freely suspended in solvent (lines 67-69). Also, we clarified that our system was a 3D microtubule-based active fluid system consisting of self-rearranging isotropic active gel, so there were no active layers like there would be in a 2D active nematic system (lines 64-67). We thank the reviewer for the suggestion, which increased the readability of the manuscript.

4. The authors don't address the concept of chaotic mixing and advective flows at all in the paper and I found that to be an omission - how does that concept relate to the measures for quantifying mixing in the submitted work? A short discussion might help put this work in context with that recent work cited as ref 24 where chaotic mixing was described for the same system. Some other groups are also considering advective flows in these systems. How would this be related to the present work?

We thank the reviewer for the suggestion on discussing the chaotic mixing and advective flows in our manuscript. We realize that our method of quantifying mixing by using the interface progression exponent, γ , and mixing time, t_0 , is not a complete characterization of the mixing dynamics of the active fluid system. Tan *et al.* characterized the chaotic mixing by introducing topological entropy and Lyapunov exponents. We considered characterizing the mixing dynamics in our system the same way, but soon we realized that it was not practical because our system is different from the one used by Tan *et al.* Their system was a 2D active nematic system in which the embedded tracers could remain in one focal plane and thus could be tracked for a long period of time, whereas our experimental system was a 3D isotropic active gel where tracers frequently moved out of the focal plane, which prevented us from continuously tracking them. If the tracers could be imaged and tracked in 3D, it would be possible to measure Lyapunov exponents and topological entropies. We are working on developing 3D imaging and tracking; but we have not fully developed the technique at this time. Nevertheless, we agree with the reviewer that measuring Lyapunov exponents and topological entropies would have provided deeper insight into the mixing dynamics of our system from the perspective of system chaotic degree. In the revised Discussion section, we address the omission of chaotic characterization and suggest potential future work characterizing the chaotic degree in the nonuniform active fluid system to gain a deeper understanding of its mixing dynamics (lines 322-325).

In response to reviewer's suggestion on exploring advective flows in our active-inactive fluid system, we adopted a dimensionless quantity, Péclet number, defined as $Pe \equiv \bar{v}_{ab} l_c / D$, where \bar{v}_{ab} represents the mean flow speed of active fluid, D is the diffusion coefficient of ATP, and l_c is the correlation length of flow velocity. The physical interpretation of this quantity is the ratio of convective transport rate to diffusive transport rate. When the Péclet number is greater than of order 1, the active transport is dominated by convection, whereas when the Péclet number is smaller than of order 1, the active transport is dominated

by diffusion. To explore how the behaviors of active transport affect the mixing of an active-inactive fluid system, we first analyzed the interface progression exponent γ as a function of Péclet number (Fig. 4b) and found that as $Pe \lesssim 3$, $\gamma \approx 1$, which corresponded to the diffusion-like mixing captured in our Fick's law-based model (Fig. 3). Interestingly, as the Péclet number is increased to $Pe \gtrsim 3$, we observed γ became greater than 1, which corresponded to the transition in interface progression to being superdiffusion-like as the convection mechanisms started to emerge. Overall, introducing the concept of advection and quantifying it with the dimensionless Péclet number allowed us to understand the progression of active-inactive interface from a more fundamental perspective of material transport. We have revised the manuscript to include the concept of advection to interpret the observed transition of interface progression throughout the manuscript (e.g., lines 146-161). We believe our manuscript is now more self-explanatory. We thank the reviewer for the suggestion that significantly improved the manuscript.

I also think that the idea that mixing is driven primarily by defects should be addressed. Can this idea be related to the present work? This was first introduced in theoretical works by Marchetti and shown experimentally in ref 24. Since the submitted paper is not the first to talk about self-mixing in active nematics, a discussion should be added to see how these different papers fit together and can be connected to the submitted work.

We thank the reviewer for pointing out a confusion in our manuscript. The work of Tan *et al.* and Marchetti (such as PRX 9, 041047 [2019]) mainly focuses on 2D active nematic systems with high nematic order in which defect dynamics play a dominant role in mixing. In contrast, our experimental system is a 3D isotropic active gel with nematic order parameter close to zero, in which mixing is mainly driven by extensile microtubule bundles. Because of this fundamental difference in mixing dynamics, we think that our work cannot be directly compared with theirs. To avoid readers misinterpreting our work as the results from a 2D active nematic system, we revised the first paragraph of Results section to clarify that our system is a 3D isotropic active gel.

With these being said, we think that it would be more elucidative to compare our work with previous research on 3D isotropic active gels. As such, we compared our works with experimental research by Sanchez *et al.* (Nature 491, 431 [2012]) and Henkin *et al.* (Philosophical Transactions. Series A, Mathematical, Physical, and Engineering Sciences 372, 20140142 [2014]) and modeling research by Varghese *et al.* (PRL 125, 268003 [2020]) and Saintillan and Shelly (Physics of Fluids 20, 123304 [2008]). We believe these comparisons better demonstrate how self-organization and mixing dynamics of 3D isotropic microtubule-kinesin active fluids with nonuniform activity are different from those with uniform activity.

5. The results presented in the paper focus on large length-scales (i.e. much larger than the active length scale. Can the authors discuss their results where the length scales approach the active length scale or even go below it? Transport measures may be different on small scales and should be ballistic-like. Can local flows on the scale of defects be tracked to get more detail on these scales?

We thank the reviewer for suggesting that we look into the smaller-scale kinematics. Indeed, our results for interface progression transitioning from diffusion-like to superdiffusion-like behaviors would be better elucidated if we could have measured the mean squared displacement (MSD) of tracers across the interface and extract the diffusion exponents (α), like previous studies by Sanchez *et al.* Nature 491, 431 (2012). We attempted to perform such experiments and analyses; however, we soon learned that it was difficult to realize because, unlike Sanchez *et al.*'s work where the active fluid had steady uniform activity, our system had a dynamic active-inactive interface whose position and width changed with time. Such a time-varying interface prevented us from collecting MSD of a tracer at a fixed activity level of the interface because the

tracer initially at the diffusion zone (low ATP concentration) of the interface may later be in the superdiffusion zone (high ATP concentration) as the interfaces passed by so it would be difficult to distinguish between the diffusive and superdiffusive data, not to mention to analyze the corresponding diffusion exponents (α). Thus, we think that investigating the tracer motions at small length scales of the active-inactive interface in our system was impractical, at least with the tools and methods we currently have.

However, we agree with the reviewer that such a small length-scale study would provide a deeper insight into the active transport of active fluid at the active-inactive interface from the perspective of microscopic kinematics. Perhaps such a characterization can be better realized in an active fluid system whose active-inactive interface does not change with time, which would make it possible to measure tracer MSD at a fixed activity level of the interface and reveal how the tracer behaviors change across the interface. Such an experimental system could be established by adopting active fluid systems that are only activated upon light exposure and become inactive when the light is turned off (such as the systems developed by Ross *et al.* Nature 572, 224 [2019] and by Zhang *et al.* Nature Materials 20, 875 [2021]) because one can use such active fluid systems to create a steady, time-independent activity gradient by applying a fixed light intensity gradient.

We thank the reviewer for the insightful suggestions. We have revised the manuscript to clarify the limitations of our studies and suggested experiments for potential future work (lines 326-338) for readers who are interested in learning about the microscopic kinematics of fluid flows at active-inactive interface.

6. It appears that Fig 3d might fit to Michaelis-Menten kinetics, it looks like the trend is approximately there. Is that trend expected for this system? Comment on the shape of the curve and prior work.

We thank the reviewer for sharing the keen observation. When we originally prepared the manuscript, we did not think the interface progression coefficient would follow the Michaelis-Menten trend because we thought it was irrelevant, but after we followed the reviewer's suggestion to fit P_I vs. C_0 to the Michaelis-Menten equation (Supplementary Fig. 4a), we found out that we were wrong because P_I vs. C_0 was well fit to the equation with goodness of fit $R^2 \geq 0.99$ (Supplementary Fig. 4b). In fact, we tried ten other flow speed-ATP relations and found that in each relation, P_I vs. C_0 followed the corresponding ATP dependence (fit curves in Supplementary Fig. 4a). This is an unexpected result; we never thought that two such unrelated ATP dependences— $P_I(C_0)$ and $\bar{v}(C)$ —were connected in our model. To explore whether there is an underlying algebra that connect P_I and C_0 via a Michaelis-Menten equation, we derived an analytical expression for the interface progression coefficient as a function of initial ATP concentration $P_I(C_0)$ (Eq. S7 in Supplementary Discussion 3), which reproduced the numerical results (see magenta curve and red dots in Fig. 3d). This expression shows that P_I was connected to C_0 via an inverse complementary error function. Coincidentally, this functional form is extremely well approximated by a Michaelis-Menten-type equation ($R^2 \geq 0.99$). We have summarized this finding in Supplementary Discussions 2 and 3 for readers who are interested in how the choice of flow speed-ATP relation $\bar{v}(C)$ affects the resulting interface progression coefficient $P_I(C_0)$. We thank the reviewer for sharing with us this interesting perspective in our interface progression coefficient that made our manuscript more inspiring.

7. In all the Figures the captions need to make it much clearer which panels are calculations and which are experimental data. The reader should not have search around for this information.

We thank the reviewer for pointing out this lack of clarity in our figure captions. To address this concern, we explicitly stated whether each figure shows experimental or simulation results in the beginning of each

figure caption. In Fig. 3, where simulations and experiments were both presented for comparison, we specified in each panel whether the data shown was simulation or experimental results.

Line 241 - You can't use "turbulences" - this should be phrased better - do you mean vortices? Areas of "active turbulence". Please clarify.

We thank the reviewer for highlighting the confusion in our manuscript. We have changed the phrase "active turbulence" to "chaotic, turbulence-like mixing flows" to improve the readability of our manuscript. We thank the reviewer for the suggestion, which has made our manuscript more understandable and clearer to a wider range of readers.

Fig S3 - the plural of spectrum is spectra.

We thank the reviewer for their correction. We have corrected the spelling mistakes in the caption of Supplementary Fig. 11.

REVIEWERS' COMMENTS

Reviewer #2 (Remarks to the Author):

In their rebuttal letter, the authors have responded in detail to all points raised in my previous report. They have modified and extended their manuscript accordingly. Thus, I support the publication of the manuscript in its present form.

Reviewer #3 (Remarks to the Author):

I am satisfied with the changes, they have greatly improved the manuscript and I recommend this work be accepted.

Reviewer 2:

In their rebuttal letter, the authors have responded in detail to all points raised in my previous report. They have modified and extended their manuscript accordingly. Thus, I support the publication of the manuscript in its present form.

We thank the reviewer for the support.

Reviewer 3:

I am satisfied with the changes, they have greatly improved the manuscript and I recommend this work be accepted.

We thank the reviewer for the recommendation.